# Diversifying Spatial-Temporal Perception for Video Domain Generalization

**Kun-Yu Lin**[1*]    **Jia-Run Du**[1*]    **Yipeng Gao**[1]    **Jiaming Zhou**[1]
**Wei-Shi Zheng**[1,2†]

[1]School of Computer Science and Engineering, Sun Yat-sen University, China
[2]Key Laboratory of Machine Intelligence and Advanced Computing, Ministry of Education, China
`{linky5,dujr6,gaoyp23,zhoujm55}@mail2.sysu.edu.cn`
`wszheng@ieee.org`

## Abstract

Video domain generalization aims to learn generalizable video classification models for unseen target domains by training in a source domain. A critical challenge of video domain generalization is to defend against the heavy reliance on domain-specific cues extracted from the source domain when recognizing target videos. To this end, we propose to perceive diverse spatial-temporal cues in videos, aiming to discover potential domain-invariant cues in addition to domain-specific cues. We contribute a novel model named Spatial-Temporal Diversification Network (STDN), which improves the diversity from both space and time dimensions of video data. First, our STDN proposes to discover various types of spatial cues within individual frames by spatial grouping. Then, our STDN proposes to explicitly model spatial-temporal dependencies between video contents at multiple space-time scales by spatial-temporal relation modeling. Extensive experiments on three benchmarks of different types demonstrate the effectiveness and versatility of our approach.

## 1 Introduction

Recently, advanced deep network architectures have achieved competitive results for video classification [1, 2, 3, 4, 5, 6, 7, 8], leading to wide applications in surveillance systems, sport analysis, health monitoring, etc [9, 10, 11]. However, existing video classification models rely on the i.i.d. assumption, *i.e.*, training and test videos are independently and identically distributed. This assumption would be easily violated, since models often face unfamiliar scenarios in real-world applications. For example, a housework robot will work in a new house, and a surveillance system will encounter illumination change caused by camera viewpoint or weather [12, 13, 14]. Holding such an assumption, the performance of video classification models would drop significantly in unfamiliar test scenarios.

To alleviate the above problem, our work studies the video domain generalization task, which aims to learn a video classification model that is generalizable in *unseen* target domains by training in a source domain [15, 16]. In this task, videos from the source and target domains follow different distributions though with an identical label space. For example, as shown in Figure 1, humans in the source domain play basketball shooting on indoor basketball courts while those in the target domain play outdoors. Different from the video domain adaptation task with available unlabeled target videos for training [17, 18, 19, 20], video domain generalization can only access the source domain during training, which is much more challenging but more practical.

---

*Equal contributions
†Corresponding author

37th Conference on Neural Information Processing Systems (NeurIPS 2023).

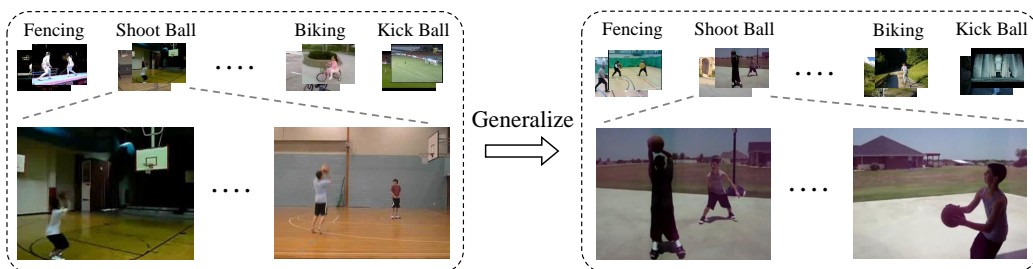

Figure 1: Video classification models suffer from the misguidance of domain-specific cues when generalizing to unseen domains. As shown in the figure, in the source domain, the static backboard provides a clearer cue compared with the blurred basketball in motion, thus prevailing video classification models are prone to recognize the class "Shoot Ball" by the backboard. However, the backboard is invisible in the target domain due to viewpoint change, and thus previous models learned in the source domain would make mistakes in recognition. Videos in the figure are from the UCF-HMDB benchmark. Best viewed in color.

A critical challenge of video domain generalization is to defend against the reliance on domain-specific cues in the source domain that are correlated with class labels. For example, as shown in Figure 1, video classification models prefer to leverage the backboard for recognizing the class "shoot ball" in the source domain, since the static backboard provides a clearer cue compared with the blurred basketball in motion (static patterns are usually easy-to-fit [21, 22, 23, 24]). However, in the target domain, the backboard is occluded due to the viewpoint, thus recognizing the class by the backboard would cause recognition errors. It is challenging to address this problem in lack of any knowledge of the target domain. Typically, existing works explore invariance across domains for learning generalizable video features [25, 15, 16]. For example, Yao et al. propose to learn generalizable temporal features by encoding information of local features into global features, assuming that local temporal features are more invariant across domains compared with global ones [15].

In this work, we propose a novel approach for video domain generalization, which explore spatial-temporal diversity in videos. Our approach aims to perceive diverse class-correlated cues from abundant video contents, and thus we would leverage not only easy-to-fit domain-specific cues but also other *potential* domain-invariant cues for recognizing videos in target domains (*e.g.*, we expect that our model can capture not only static backboards but also dynamic basketballs in the source domain). As a result, our approach can alleviate the overfitting of domain-specific cues in the source domain and generalize better in target domains by leveraging those potential domain-invariant cues. Specifically, we propose to explore the diversity from both space and time dimensions of video data, leading to a novel architecture named Spatial-Temporal Diversification Network (STDN). Our contributions are summarized as follows:

- We propose Spatial Grouping Module to discover various groups of spatial cues within individual frames by embedding a clustering-like process, enriching the diversity from a spatial modeling perspective.

- We propose Spatial-Temporal Relation Module to explicitly model spatial-temporal dependencies between video contents at multiple space-time scales, enriching the diversity from a spatial-temporal relation modeling perspective.

- Extensive experiments are conducted on three benchmarks of different types, including two newly designed benchmarks, and the results demonstrate the effectiveness and versatility of our proposed method.

## 2 Related Works

**Video Classification** aims to recognize actions or events in videos. Recently, many advanced deep learning architectures have been proposed for video classification. 3D CNNs extend the 2D convolution to 3D convolution for video feature learning [1, 2, 3, 26, 27, 28, 29, 30]. Another

type of models first applies 2D convolution for frame-level spatial modeling and then conducts temporal modeling based on frame features [5, 6, 31, 32, 33]. Some works propose to couple explicit shifts along the time dimension for efficient temporal modeling [7, 34, 35]. More recently, pioneer works have made efforts in adapting Vision Transformer [36] for video classification [37, 38, 39, 40, 41, 42, 43, 44, 45]. Although these advanced architectures achieve appealing performance, they usually assume an identical test distribution to the training one, which is not practical in real-world applications.

**Video Domain Generalization** aims to train video classification models in a source domain for generalizing to *unseen* target domains. With target videos inaccessible during training, existing works usually assume different types of invariance across domains to defend against the reliance on domain-specific cues [25, 15, 16]. For example, Yao et al. propose to learn generalizable temporal features according to an assumption from empirical findings, *i.e.*, local temporal features are more invariant across domains compared with the global ones [15]; Planamente et al. propose to constrain a consistency across visual and audio modalities by relative norm alignment for addressing domain generalization of egocentric action recognition [16]. In this work, we propose to perceive diverse class-correlated spatial-temporal cues in source videos, which alleviates the misguidance of domain-specific cues in a way that is orthogonal to previous works.

**Video Domain Adaptation** aims to learn transferable video classification models for a label-free target domain by transferring knowledge from a label-sufficient source domain [17, 18]. Different from video domain generalization, video domain adaptation is oriented to a specific *seen* unlabeled target domain. Typically, existing works learn invariance between labeled source videos and unlabeled target videos to tackle video domain adaptation. A class of representative works propose to learn domain-invariant temporal features by designing temporal modeling modules [18, 19, 46, 47]. In addition, Choi et al. [20, 48] propose self-supervised methods adaptive to video data. Furthermore, multi-modal works explore information interaction between different modalities (*e.g.*, RGB, Flow, Audio) for domain-invariant feature learning [49, 50, 51, 52, 53].

**General Domain Generalization**, also known as out-of-distribution generalization, studies learning models generalizable to out-of-distribution data for the image classification task . In recent years, a plethora of methods have been proposed to address domain generalization [25, 54, 55, 56]. Prevailing methods are mainly based on feature alignment [57, 58, 59], domain adversarial learning [60, 61], invariant risk minimization [62, 63, 64, 65], meta learning [66, 67, 68, 69, 70], data augmentation [71, 72, 73, 74, 75], etc. In addition, ensemble learning is an effective way to learn generalizable models [76, 77, 78]. And recently, Zhu et al. develop a theory showing that ensemble learning can provably improve test accuracy by discovering the "multi-view" structure of data [79], which partially inspires our approach. Among architecture-based methods [80, 81], Meng et al. propose to redesign attention modules for learning diverse task-related features [80]. Different from existing general domain generalization methods, we propose a domain generalization method specific to video classification, which explores diverse class-correlated information in intrinsic space and time dimensions of video data. There are some works that study the identification of out-of-distribution data of different categories from training data [82, 83, 84, 85, 86, 87, 88], but this topic is not within the scope of our work.

## 3 Spatial-Temporal Diversification Network

In this section, we illustrate our proposed Spatial-Temporal Diversification Network (STDN) in detail, which perceives diverse class-correlated spatial-temporal cues from video contents for generalization in unseen target domains.

### 3.1 Problem Formulation

In video domain generalization, a set of labeled videos $\mathcal{D} = \{(x, y)\}$ from a source domain are given for training, where $x \in \mathcal{X}$ and $y \in \mathcal{Y}$ denote a source video and its corresponding class label. Given only the source video set, the goal of video domain generalization is to learn a video classification model that is generalizable in *unseen* target domains. The source and target domains follow different but related distributions with the same label space $\mathcal{Y} = \{0, 1, \ldots, C - 1\}$, where $C$ denotes the number of video classes. Following the standard video domain generalization setting [15], each video is evenly divided into $N$ segments, and one frame is sampled from each segment as the model input

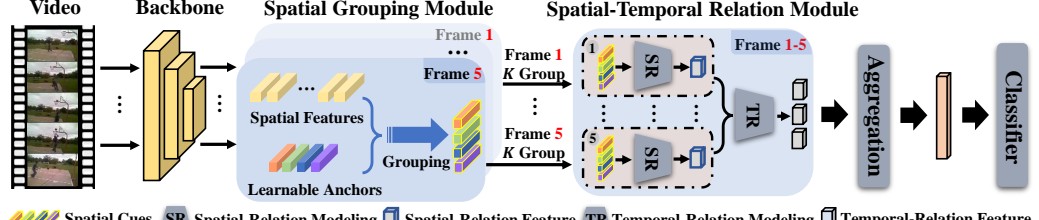

Figure 2: An overview of our proposed Spatial-Temporal Diversification Network (STDN). We use a video of $N = 5$ segments with $K = 4$ spatial groups for example. After backbone feature extraction, our STDN extracts spatial cues of $K$ types for each frame by the Spatial Grouping Module, enriching the diversity in spatial modeling. Then, our STDN explicitly models spatial-temporal dependencies at multiple space-time scales, enriching the diversity in spatial-temporal relation modeling. Best viewed in color.

during training and testing, *i.e.*, $x = \{x_1, x_2, \ldots, x_N\}$ and $x_n$ denotes the $n$-th sampled frame from the $n$-th segment.

## 3.2 Model Overview

Aiming at generalization in unseen target domains, our idea is to perceive rich and diverse class-correlated cues in the source domain. In this way, our model would leverage not only easy-to-fit domain-specific cues but also other potential domain-invariant cues for recognizing videos in the target domain, alleviating the misguidance of domain-specific cues. Considering the intrinsic space and time dimensions of video data, we propose to explore the diversity in both spatial and temporal modeling. An overview of our proposed STDN is shown in Figure 2. Firstly, given the video $x$, our STDN takes $N$ sampled frames as input and separately extracts $N$ spatial feature maps $\{z_1, z_2, \ldots, z_N\}$ by the backbone (*e.g.*, ResNet [89]), where $z_n \in \mathbb{R}^{H \times W \times D}$ denotes the feature map of the $n$-th frame, $D$ denotes the feature dimension and $H \times W$ denotes the size of feature maps. Then, we extract spatial cues of $K$ types (groups) from each spatial feature map by our proposed Spatial Grouping Module, aiming to enrich the spatial diversity. In the Spatial Grouping Module, two entropy-based losses are introduced to enhance the distinction between different spatial cues. On top of the Spatial Grouping Module, we propose to explicitly model spatial-temporal dependencies between video contents at multiple space-time scales by our proposed Spatial-Temporal Relation Module. The learning of the Spatial-Temporal Relation Module is guided by a relation discrimination loss, which ensures the diversity of the extracted spatial-temporal relation features. Finally, diverse spatial-temporal features are aggregated for video domain generalization.

## 3.3 Spatial Grouping Module

Our proposed Spatial Grouping Module aims to discover diverse class-correlated spatial cues from abundant contents of individual frames, which enriches the diversity from a spatial modeling perspective for video domain generalization. Our Spatial Grouping Module extracts various spatial cues of different types by partitioning features from different spatial positions into several groups within individual frames. In this way, our Spatial Grouping Module discovers more diverse spatial cues, compared with prevailing approaches that extract an integrated feature for each frame (*e.g.*, by average pooling).

As shown in Figure 3 (a), given the spatial feature map $z_n \in \mathbb{R}^{H \times W \times D}$ of the $n$-th frame, our proposed Spatial Grouping Module learns to extract $K$ spatial cues by aggregating the $HW$ spatial features. Specifically, the proposed spatial grouping process is conducted based on $K$ learnable anchor features $\{a_{n,1}, a_{n,2}, \ldots, a_{n,K}\}$, where $a_{n,k} \in \mathbb{R}^D$ denotes the anchor feature of the $k$-th spatial group for the $n$-th frame. Then, we calculate the probability of assigning a spatial feature to each spatial group, which is formulated as follows:

$$p_{n,i,k} = \frac{\exp\left(-\text{dist}\left(z_{n,i}, a_{n,k}\right)/\tau\right)}{\sum_{j=1}^{K} \exp\left(-\text{dist}\left(z_{n,i}, a_{n,j}\right)/\tau\right)}, \tag{1}$$

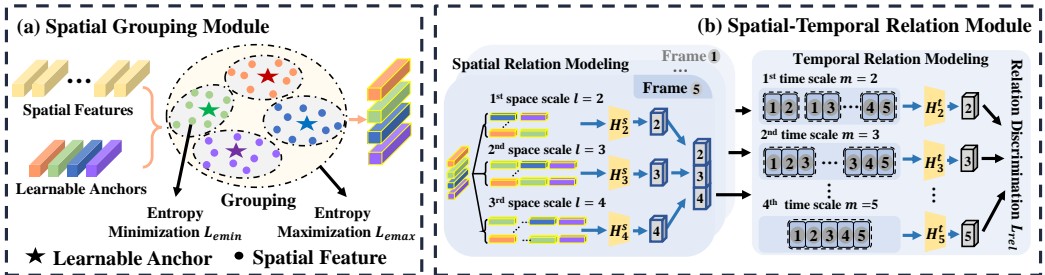

Figure 3: Overviews of our proposed (a) Spatial Grouping Module and (b) Spatial-Temporal Relation Module. Best viewed in color.

where $z_{n,i} \in \mathbb{R}^D$ denotes the $i$-th spatial feature in the feature map $z_n$ ($i \in [1, 2, \ldots, HW]$), $\text{dist}(\cdot, \cdot)$ denotes the Euclidean distance metric and $\tau$ is the temperature factor. According to Eq. (1), if the spatial feature $z_{n,i}$ is closer to the anchor feature $a_{n,k}$, then the $z_{n,i}$ will be assigned to the $k$-th spatial group with higher probability. After group partition, our Spatial Grouping Module produces $K$ integrated features representing $K$ different spatial cues by aggregating spatial features in each group. The integration process is formulated as follows:

$$z_{n,k}^s = \frac{1}{\sum_{i=1}^{HW} p_{n,i,k}} \sum_{i=1}^{HW} p_{n,i,k} * z_{n,i},$$ (2)

where $z_{n,k}^s$ denotes the spatial cues integrated from the $k$-th group within the $n$-th frame.

In order to extract spatial cues of diverse types, we introduce two entropy-based losses to enhance the distinction between different spatial groups. The first one is an entropy minimization loss to enhance the confidence of group assignment for each spatial feature. The loss is formulated as follows:

$$L_{\text{emin}} = -\frac{1}{NHW} \sum_{n=1}^{N} \sum_{i=1}^{HW} \sum_{k=1}^{K} p_{n,i,k} \log\left(p_{n,i,k}\right).$$ (3)

For the assignment probability vector $p_{n,i} = [p_{n,i,1}, p_{n,i,2}, \ldots, p_{n,i,K}]^T \in \mathbb{R}^{K \times D}$, if the entropy is minimized, then the feature $z_{n,i}$ will be confidently assigned to a specific spatial group. The second loss is an entropy maximization loss for the mean assignment probability vector, which guarantees that those $HW$ spatial features are assigned to different spatial groups. Specifically, the loss is formulated as follows:

$$L_{\text{emax}} = \frac{1}{N} \sum_{n=1}^{N} \sum_{k=1}^{K} \bar{p}_{n,k} \log\left(\bar{p}_{n,k}\right),$$ (4)

where $\bar{p}_{n,k} = \frac{1}{HW} \sum_{i=1}^{HW} p_{n,i,k}$ denotes the mean probability of assigning features to the $k$-th group within the $n$-th frame. For the mean assignment probability vector $\bar{p}_n = [\bar{p}_{n,1}, \bar{p}_{n,2}, \ldots, \bar{p}_{n,K}]^T \in \mathbb{R}^{K \times D}$, if the entropy is maximized, then the spatial features $\{z_{n,i}\}$ will be uniformly assigned to $K$ spatial groups. By using the two entropy-based losses, we guarantee that spatial features are different from each other across different spatial groups, enriching the diversity of extracted spatial cues.

In the Spatial Grouping Module, the learnable anchor feature for each group is extracted by weighted combining those $HW$ spatial features, and the weights are calculated conditioned on the feature map $z_n$ by using a lightweight two-layer convolutional network. In this way, the spatial grouping process can be regarded as conducting clustering over spatial features within individual frames. All involved parameters in the module are end-to-end trained together with the main network, *i.e.*, we contribute a parametric clustering module to group spatial features for improving the spatial diversity.

### 3.4 Spatial-Temporal Relation Module

Our proposed Spatial-Temporal Relation Module aims to discover diverse class-correlated spatial-temporal cues from abundant video contents, which enriches the diversity from a spatial-temporal relation modeling perspective for video domain generalization. As demonstrated by previous works [4, 6, 90], there are rich dependencies between entities over space and time dimensions in videos, which

is crucial for video classification. Accordingly, we propose to explicitly model spatial-temporal dependencies between video cues at multiple space-time scales. Our proposed Spatial-Temporal Relation Module conducts dependency modeling at space and time dimensions separately, and an overview of the module is shown in Figure 3 (b).

First, based on the spatial cues extracted by our Spatial Grouping Module, we conduct spatial dependency modeling between these spatial cues at multiple space scales. Specifically, given the representations of spatial cues $z_n^s = [z_{n,1}^s, z_{n,2}^s, \ldots, z_{n,K}^s]^T \in \mathbb{R}^{K \times D}$ for the $n$-th frame, we extract the spatial relation feature at the $l$-th space scale by the spatial dependency modeling function $R_l^s(\cdot)$ as follows:

$$R_l^s(z_n^s) = \mathbb{E}_{k_1, k_2, \ldots, k_l} \left[ H_l^s(z_{n,k_1}^s, z_{n,k_2}^s, \ldots, z_{n,k_l}^s) \right] \in \mathbb{R}^{D_s}, \tag{5}$$

where $\mathbb{E}[\cdot]$ denotes the expectation calculation and $H_l^s(\cdot, \cdot, \ldots, \cdot)$ denotes a linear projection function after feature concatenation. The index set $\{k_1, k_2, \ldots, k_l\}$ denotes the index of spatial features uniformly sampled from the $K$ spatial features, where the index $l \in \{2, 3, \ldots, K\}$ indicates the space scale, $k_1 \neq k_2 \neq \cdots \neq k_l$ and $k_i \in \{1, 2, \ldots, K\}$. For each frame, we extract $K - 1$ spatial relation features by dependency modeling at $K - 1$ space scales separately. And, we concatenate these spatial relation features and produce an integrated feature for each frame, which is given by $\hat{z}_n = [R_2^s(z_n^s)^T, R_3^s(z_n^s)^T, \ldots, R_K^s(z_n^s)^T, G(z_n)^T]^T \in \mathbb{R}^{K D_s}$. In the integrated feature $\hat{z}_n$, $G(z_n) \in \mathbb{R}^{D_s}$ denotes the global feature extracted from the feature map $z_n$ by a convolution layer.

Second, based on the frame-level integrated features, we conduct temporal dependency modeling between frames at multiple time scales. Specifically, given $N$ frame-level features denoted by $\hat{z} = [\hat{z}_1, \hat{z}_2, \ldots, \hat{z}_N]$, we extract the temporal relation feature at the $m$-th time scale by the temporal dependency modeling function $R_m^t(\cdot)$ as follows:

$$R_m^t(\hat{z}) = \mathbb{E}_{n_1 < n_2 < \cdots < n_m} \left[ H_m^t(\hat{z}_{n_1}, \hat{z}_{n_2}, \ldots, \hat{z}_{n_m}) \right] \in \mathbb{R}^{D_t}, \tag{6}$$

where $H_m^t(\cdot, \cdot, \ldots, \cdot)$ denotes a linear projection function after feature concatenation. The index set $\{n_1, n_2, \ldots, n_m\}$ denotes the index of frame features randomly sampled from the $N$ frame features, where the index $m \in \{2, 3, \ldots, N\}$ indicates the time scale and $n_i \in \{1, 2, \ldots, N\}$. Note that we keep the relative order of sampled frames for temporal modeling. By using $N - 1$ temporal dependency modeling functions, we extract $N - 1$ temporal relation features at $N - 1$ time scales for each video.

To ensure the diversity of temporal relation features, we propose a relation discrimination loss to constrain that different temporal dependency modeling functions (*i.e.*, different time scales) capture different temporal cues. This loss constrains that a relation classifier can distinguish one relation feature from not only relation features of other classes but also relation features of the same class but of other time scales. Thus, it avoids the feature collapse of learned temporal relation features. Specifically, the loss is formulated as follows:

$$L_{\text{rel}} = \frac{1}{N - 1} \sum_{m=2}^{N} \text{CE}(F_{\text{rel}}(\tilde{z}_m), \tilde{y}_m), \tag{7}$$

where $\tilde{z}_m = R_m^t(\hat{z})$ denotes the temporal relation feature at the $m$-th time scale, $F_{\text{rel}}(\cdot)$ denotes a relation classifier that classifying $(N - 1) * C$ classes, and $\text{CE}(\cdot, \cdot)$ denotes the cross-entropy loss. The $\tilde{y}_m$ denotes the relation label of the video $x$ with label $y$, *i.e.*, $\tilde{y}_m = y * (N - 1) + (m - 2) \in \{0, 1, 2, \ldots, (N - 1) * C - 1\}$. In this way, the loss forces different temporal dependency modeling functions to capture different class-correlated temporal cues in the video since the captured temporal cues are discriminative across scales. By incorporating the Spatial-Temporal Relation Module with the relation discrimination loss $L_{\text{rel}}$, we extract rich and diverse spatial-temporal cues.

**Feature Aggregation:** After exploring spatial-temporal diversity by our proposed Spatial Grouping Module and Spatial-Temporal Relation Module, our model discovers diverse class-correlated spatial-temporal cues from abundant video contents. Then, we aggregate these diverse spatial-temporal features for video classification. Specifically, the feature aggregation is formulated as $\check{z} = \sum_{m=2}^{N} H_m^a(\tilde{z}_m)$, where $H_m^a(\cdot)$ denotes a small SE-based block [91] for modulating the $m$-th temporal relation features.

**Overall Training and Test:** We adopt a video classification loss on top of the aggregated feature $\check{z}$ given by $L_{\text{cls}} = \text{CE}(F(\check{z}), y)$, where $F(\cdot)$ is a video classifier. Overall, the training loss of our

Table 1: Comparison with state-of-the-art methods on the UCF-HMDB benchmark. **Red** and blue denotes the best and second best. Results of all compared methods are from VideoDG [15].

| Arch | DG Method | UCF→HMDB | HMDB→UCF | Arch | DG Method | UCF→HMDB | HMDB→UCF |
|---|---|---|---|---|---|---|---|
| TSN [5] | ERM | $51.4_{\pm0.2}$ | $68.6_{\pm0.3}$ | TSM [7] | ERM | $52.2_{\pm0.3}$ | $69.2_{\pm0.3}$ |
| | $ADA_{sem}$ [71] | $51.1_{\pm0.3}$ | $68.2_{\pm0.2}$ | | $ADA_{sem}$[71] | $51.3_{\pm0.3}$ | $68.6_{\pm0.3}$ |
| | $ADA_{pixel}$ [71] | $49.6_{\pm0.3}$ | $67.4_{\pm0.2}$ | | $ADA_{pixel}$ [71] | $52.7_{\pm0.3}$ | $68.3_{\pm0.2}$ |
| | M-ADA [92] | $52.4_{\pm0.2}$ | $69.2_{\pm0.2}$ | | M-ADA [92] | $52.5_{\pm0.2}$ | $69.1_{\pm0.3}$ |
| | Jigsaw [93] | $51.5_{\pm0.3}$ | $68.5_{\pm0.3}$ | | Jigsaw [93] | $52.5_{\pm0.3}$ | $68.9_{\pm0.3}$ |
| APN [15] | ERM | $54.3_{\pm0.3}$ | $71.4_{\pm0.3}$ | TRN [6] | ERM | $52.4_{\pm0.3}$ | $69.8_{\pm0.3}$ |
| | $ADA_{sem}$ [71] | $55.2_{\pm0.3}$ | $71.9_{\pm0.3}$ | | $ADA_{sem}$ [71] | $52.8_{\pm0.2}$ | $69.6_{\pm0.5}$ |
| | $ADA_{pixel}$ [71] | $56.9_{\pm0.2}$ | $72.2_{\pm0.3}$ | | $ADA_{pixel}$ [71] | $52.1_{\pm0.3}$ | $70.6_{\pm0.2}$ |
| | M-ADA [92] | $55.6_{\pm0.3}$ | $71.5_{\pm0.3}$ | | M-ADA [92] | $53.4_{\pm0.3}$ | $69.9_{\pm0.3}$ |
| | Jigsaw [93] | $55.2_{\pm0.4}$ | $72.4_{\pm0.3}$ | | Jigsaw [93] | $53.3_{\pm0.3}$ | $70.1_{\pm0.3}$ |
| VideoDG [15] | | $59.1_{\pm0.3}$ | $74.9_{\pm0.3}$ | STDN (Ours) | | $60.2_{\pm0.5}$ | $77.1_{\pm0.4}$ |

proposed STDN is given as follows:

$$L = L_{\mathrm{cls}} + \lambda_{\mathrm{ent}}L_{\mathrm{emin}} + \lambda_{\mathrm{ent}}L_{\mathrm{emax}} + \lambda_{\mathrm{rel}}L_{\mathrm{rel}}, \qquad (8)$$

where $\lambda_{\mathrm{ent}}$ and $\lambda_{\mathrm{rel}}$ are hyperparameters for trade-off. Following the standard protocol [15], we use source videos for training and test the model on target videos for evaluation.

## 4 Experiments

### 4.1 Benchmarks and Experimental Setups

To demonstrate the effectiveness and versatility of our proposed Spatial-Temporal Diversification Network (STDN), we adopt three benchmarks of different types for experiments, including two newly designed benchmarks, namely EPIC-Kitchens-DG and Jester-DG. For these two new benchmarks, we select video categories and construct domains following previous video domain adaptation works [49, 19]. We split the source video set into training and validation sets following previous source validation protocols [25, 15], *i.e.*, a reasonable in-domain model selection scheme for better generalization ability in unseen target domains. We reproduce general domain generalization methods (cooperated with video classification architectures) and state-of-the-art video domain generalization methods for comparison. We report mean and standard deviation of accuracy over three random trials for all methods.

**UCF-HMDB** is the most widely used benchmark for cross-domain video classification [15, 18], which contains 3,809 videos of 12 overlapping sport categories shared by UCF101 [94] and HMDB51 [95]. The videos in UCF101 are mostly captured from certain scenarios or similar environments, and the videos in HMDB51 are captured from unconstrained environments and different camera viewpoints. This benchmark includes two subtasks, i.e., UCF→HMDB and HMDB→UCF.

**EPIC-Kitchens-DG** is a *cross-scene egocentric action recognition* benchmark, which consists of 10,094 videos across 8 egocentric action classes from three domains (scenes), following Munro et al. [49]. The three domains of EPIC-Kitchens-DG (*i.e.*, D1, D2, D3) correspond to three largest kitchens (*i.e.*, P08, P01, P22) from the large-scale egocentric action recognition dataset EPIC-Kitchens-55 [96]. This benchmark includes six subtasks constructed from three domains.

**Jester-DG** is a *cross-domain hand gesture recognition* benchmark. We select videos from the Jester dataset [97] and construct two domains following Pan et al. [19]. The source (S) and target (T) domains contain 51,498 and 51,415 video clips across 7 categories, respectively. The videos in EPIC-Kitchens-DG and Jester-DG benchmarks are both hand-centric, but they are captured from different views, namely first-person and third-person views.

**Implementation details:** We use ResNet50 [89] pretrained on ImageNet [98] as the backbone for frame-level feature extraction following the standard video domain generalization protocol [15]. The backbone takes frames of size $224 \times 224$ as input and outputs feature maps of size $7 \times 7 \times 2048$. We take $N = 5$ frames for each video for temporal modeling. We set $K = 4$, $\tau = 0.5$, $D_s = 192$ and $D_t = 256$. $F(\cdot)$ is a linear classifier and $F_{\mathrm{rel}}(\cdot)$ is an MLP classifier. All parameters are optimized using mini-batch SGD with a batch size of 32, a momentum of 0.9, a learning rate of

Table 2: Comparison with state-of-the-art methods on the EPIC-Kitchens-DG and Jester-DG benchmarks. **Red** and blue denotes the best and second best. Results of all compared methods are reproduced following their official implementations.

| Arch | DG Method | Epic-Kitchens-DG | | | | | | | Jester-DG |
|------|-----------|---------|---------|---------|---------|---------|---------|---------|-----------|
| | | D1→D2 | D1→D3 | D2→D1 | D2→D3 | D3→D1 | D3→D2 | Average | S→T |
| TSN [5] | ERM | $33.6_{\pm0.6}$ | $31.3_{\pm0.5}$ | $32.5_{\pm0.9}$ | $36.1_{\pm1.2}$ | $31.7_{\pm0.5}$ | $40.2_{\pm0.7}$ | $34.2_{\pm0.4}$ | $47.5_{\pm0.7}$ |
| | Mixup [72] | $33.6_{\pm0.2}$ | $29.1_{\pm0.3}$ | $31.2_{\pm0.7}$ | $36.5_{\pm0.4}$ | $33.0_{\pm0.5}$ | $39.7_{\pm0.1}$ | $33.9_{\pm0.3}$ | $47.8_{\pm0.5}$ |
| | IRM [62] | $34.6_{\pm0.1}$ | $30.3_{\pm1.3}$ | $31.2_{\pm1.2}$ | $36.3_{\pm0.2}$ | $32.3_{\pm0.1}$ | $40.0_{\pm0.3}$ | $34.1_{\pm0.5}$ | $47.6_{\pm0.5}$ |
| | ADA [71] | $33.8_{\pm0.2}$ | $30.3_{\pm0.9}$ | $31.1_{\pm1.4}$ | $36.4_{\pm0.6}$ | $33.3_{\pm0.4}$ | $40.1_{\pm0.8}$ | $34.2_{\pm0.6}$ | $47.6_{\pm0.3}$ |
| | COP [100] | $34.7_{\pm0.1}$ | $29.7_{\pm1.6}$ | $31.5_{\pm0.8}$ | $36.7_{\pm0.5}$ | $31.5_{\pm0.1}$ | $40.3_{\pm0.5}$ | $34.1_{\pm0.6}$ | $47.3_{\pm0.9}$ |
| TSM [7] | ERM | $34.6_{\pm0.5}$ | $32.3_{\pm1.2}$ | $30.2_{\pm0.9}$ | $34.8_{\pm0.7}$ | $31.2_{\pm1.9}$ | $39.9_{\pm1.5}$ | $33.8_{\pm0.6}$ | $47.1_{\pm0.4}$ |
| | Mixup [72] | $34.1_{\pm0.7}$ | $29.3_{\pm0.6}$ | $28.1_{\pm0.2}$ | $32.4_{\pm0.3}$ | $31.7_{\pm0.7}$ | $39.1_{\pm0.1}$ | $32.4_{\pm0.3}$ | $47.3_{\pm0.4}$ |
| | IRM [62] | $34.3_{\pm0.2}$ | $29.3_{\pm1.5}$ | $31.1_{\pm0.8}$ | $36.3_{\pm0.7}$ | $31.6_{\pm1.0}$ | $38.5_{\pm0.5}$ | $33.5_{\pm0.7}$ | $46.8_{\pm0.7}$ |
| | ADA [71] | $34.3_{\pm0.8}$ | $30.6_{\pm1.0}$ | $30.0_{\pm1.0}$ | $34.6_{\pm1.3}$ | $31.8_{\pm0.6}$ | $38.8_{\pm0.9}$ | $33.4_{\pm0.6}$ | $47.1_{\pm0.5}$ |
| | COP [100] | $34.9_{\pm0.3}$ | $32.1_{\pm1.1}$ | $30.5_{\pm0.2}$ | $32.2_{\pm0.6}$ | $31.8_{\pm1.5}$ | $37.5_{\pm0.9}$ | $33.2_{\pm0.4}$ | $46.4_{\pm0.6}$ |
| APN [15] | ERM | $35.2_{\pm0.8}$ | $30.2_{\pm2.1}$ | $35.3_{\pm1.0}$ | $43.2_{\pm1.2}$ | $36.4_{\pm0.3}$ | $45.3_{\pm0.9}$ | $37.6_{\pm0.7}$ | $46.9_{\pm0.5}$ |
| | Mixup [72] | $34.7_{\pm0.2}$ | $30.9_{\pm0.7}$ | $33.8_{\pm0.7}$ | $44.7_{\pm0.5}$ | $36.4_{\pm0.3}$ | $44.4_{\pm0.7}$ | $37.5_{\pm0.4}$ | $47.8_{\pm0.1}$ |
| | IRM [62] | $33.6_{\pm0.5}$ | $28.5_{\pm1.6}$ | $34.3_{\pm0.4}$ | $42.8_{\pm0.8}$ | $34.6_{\pm0.6}$ | $44.0_{\pm0.7}$ | $36.3_{\pm0.5}$ | $47.0_{\pm0.3}$ |
| | ADA [71] | $36.0_{\pm0.3}$ | $29.5_{\pm0.2}$ | $35.0_{\pm0.4}$ | $43.0_{\pm0.3}$ | $37.6_{\pm0.4}$ | $45.6_{\pm2.3}$ | $37.8_{\pm0.6}$ | $47.8_{\pm0.4}$ |
| | COP [100] | $36.5_{\pm0.8}$ | $32.1_{\pm0.7}$ | $37.6_{\pm1.8}$ | $41.8_{\pm0.8}$ | $37.9_{\pm0.2}$ | $49.9_{\pm1.3}$ | $39.3_{\pm0.6}$ | $47.2_{\pm0.3}$ |
| TRN [6] | ERM | $36.8_{\pm1.4}$ | $32.1_{\pm1.2}$ | $34.2_{\pm1.0}$ | $44.7_{\pm0.5}$ | $37.6_{\pm0.9}$ | $48.9_{\pm0.1}$ | $39.1_{\pm0.7}$ | $46.2_{\pm0.4}$ |
| | Mixup [72] | $37.5_{\pm1.0}$ | $31.2_{\pm1.6}$ | $35.3_{\pm1.3}$ | $43.2_{\pm1.4}$ | $39.0_{\pm0.6}$ | $48.1_{\pm0.2}$ | $39.0_{\pm0.6}$ | $46.7_{\pm0.1}$ |
| | IRM [62] | $37.8_{\pm2.3}$ | $30.5_{\pm0.6}$ | $37.0_{\pm2.6}$ | $42.6_{\pm1.3}$ | $40.3_{\pm0.5}$ | $47.9_{\pm0.6}$ | $39.3_{\pm1.1}$ | $46.5_{\pm0.5}$ |
| | ADA [71] | $38.4_{\pm1.0}$ | $30.4_{\pm1.2}$ | $35.9_{\pm1.3}$ | $41.2_{\pm0.6}$ | $38.8_{\pm0.5}$ | $47.5_{\pm1.1}$ | $38.7_{\pm0.8}$ | $46.1_{\pm0.4}$ |
| | COP [100] | $36.8_{\pm1.1}$ | $34.2_{\pm2.7}$ | $35.8_{\pm1.1}$ | $39.6_{\pm2.4}$ | $38.1_{\pm0.5}$ | $49.0_{\pm2.0}$ | $38.9_{\pm1.1}$ | $47.8_{\pm0.5}$ |
| VideoDG [15] | | $36.2_{\pm0.4}$ | $31.9_{\pm0.2}$ | $36.5_{\pm0.5}$ | $40.5_{\pm0.8}$ | $39.5_{\pm0.5}$ | $49.1_{\pm0.7}$ | $39.0_{\pm0.6}$ | $47.5_{\pm0.1}$ |
| STDN (Ours) | | $40.5_{\pm0.5}$ | $38.6_{\pm0.1}$ | $38.5_{\pm2.6}$ | $44.0_{\pm1.3}$ | $40.4_{\pm1.6}$ | $47.2_{\pm1.2}$ | $\mathbf{41.6_{\pm1.0}}$ | $\mathbf{49.8_{\pm0.4}}$ |

1e-3 and a weight decay of 5e-4. By default, the trade-off hyperparameters are set as $\lambda_{\mathrm{ent}} = 0.1$ and $\lambda_{\mathrm{rel}} = 0.5$. We adopt an efficient feature augmentation technique namely MixStyle [74] for simulating novel target domains during training. We modify the original MixStyle to adapt the video data, *i.e.*, we calculate the mean and standard deviation across both space and time dimensions within each channel of each instance (instead of only space dimension for image data). All experiments are conducted by PyTorch [99] with four NVIDIA GTX 1080Ti GPUs. The code is released at https://github.com/KunyuLin/STDN/.

## 4.2 Results

**Comparison with State-of-the-arts:** We compare our proposed STDN with two types of state-of-the-arts: 1) general domain generalization (DG) methods cooperated with different video classification architectures; 2) state-of-the-art video domain generalization methods. For the two newly designed benchmarks (*i.e.*, Epic-Kitchens-DG and Jester-DG), we adopt five different types of general domain generalization methods for comparison, including Empirical Risk Minimization (ERM), Mixup [72], Invariant Risk Minimization (IRM) [62], Adversarial Data Augmentation (ADA) [71], Clip Order Prediction (COP) [100]. All results are summarized in Table 1 and 2. On all the three benchmarks, our STDN outperforms all the state-of-the-art methods. Specifically, our STDN obtains performance improvement by 2.2% and 2.3% on HMDB→UCF and Epic-Kitchens-DG respectively, which is significant compared with previous state-of-the-arts. In addition, VideoDG [15] obtains lower performance than their proposed architecture APN [15] on Epic-Kitchens-DG and Jester-DG. By contrast, our superiority on three benchmarks of different types verifies the effectiveness and versatility of our proposed STDN, demonstrating the effectiveness of perceiving diverse spatial-temporal cues. In the supplemental material, we make an attempt to compare with two variants of Planamente et al. [16] to show our effectiveness, following some works in the domain adaptation field [101, 102, 103].

**Ablation Study:** We analyze the effects of each component in our proposed STDN, as shown in Table 3. Following the training scheme of TSN [5], we apply a classifier on top of the backbone as our baseline. By stacking our proposed Spatial Grouping Module (SGM) on top of the backbone, we obtain significant improvement over the baseline (*i.e.*, 2.2% on UCF→HMDB and 2.0% on HMDB→UCF), demonstrating the effectiveness of extracting different types of spatial cues within individual frames. Then, by introducing the temporal de-

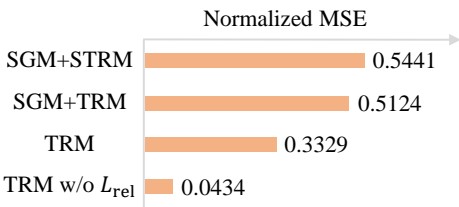

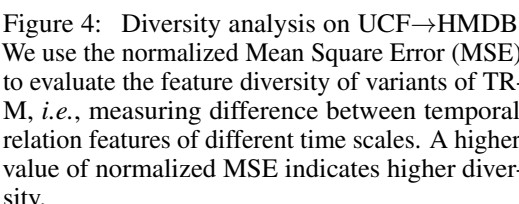

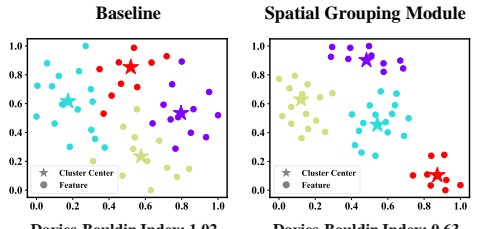

Figure 4: Diversity analysis on UCF→HMDB. We use the normalized Mean Square Error (MSE) to evaluate the feature diversity of variants of TR-M, *i.e.*, measuring difference between temporal relation features of different time scales. A higher value of normalized MSE indicates higher diversity.

Figure 5: T-SNE visualization of spatial features. For both the baseline and SGM, we cluster the set of spatial features into $K = 4$ clusters by $K$-means before visualization. In the figure, dots stand for spatial feature vectors and stars stand for cluster centers, and different colors denote different clusters.

pendency modeling of our Spatial-Temporal Relation Module (denoted by TRM), we obtain 1.8% and 1.4% improvement on UCF→HMDB and HMDB→UCF, respectively. It should be noted that the relation discrimination loss $L_{rel}$ is an important part in temporal dependency

modeling, since we obtain very minor performance improvement without the loss in temporal dependency modeling. By introducing the full Spatial-Temporal Relation Module (STR-M), we obtain 3.4% and 2.3% improvement over "Backbone+SGM" on UCF→HMDB and HMDB→UCF, respectively. These results demonstrate the effectiveness of modeling dependencies between various video cues in both

Table 3: Ablation study on UCF-HMDB.

| Method | UCF→HMDB | HMDB→UCF |
|---|---|---|
| Backbone | $52.7_{\pm 0.3}$ | $71.9_{\pm 0.3}$ |
| +SGM | $54.9_{\pm 0.3}$ | $73.9_{\pm 0.4}$ |
| +TRM | $56.7_{\pm 0.2}$ | $75.3_{\pm 0.4}$ |
| +STRM | $58.3_{\pm 0.4}$ | $76.2_{\pm 0.3}$ |
| +MixStyle | $59.3_{\pm 0.3}$ | $76.6_{\pm 0.2}$ |
| Full STDN | $60.2_{\pm 0.5}$ | $77.1_{\pm 0.4}$ |

space and time dimensions, which enriches the diversity in spatial-temporal relation modeling. Moreover, by introducing the feature augmentation technique MixStyle, we obtain further improvement. Finally, our full model aggregates diverse spatial-temporal features, leading to better generalization performance on both UCF→HMDB and HMDB→UCF.

**Diversity Analysis:** We make a quantitative analysis to the diversity of learned video features for our model. Specifically, we evaluate the difference between temporal relation features of different time scales, measured by the normalized Mean Square Error (MSE) between feature vectors. A higher value of normalized MSE indicates a large difference. As shown in Figure 4, without our relation discrimination loss $L_{rel}$, learned temporal relation features at different time scales hold very small difference (implying feature collapse). By introducing $L_{rel}$, our TRM improves the diversity, indicated by the higher MSE value. By introducing our Spatial Grouping Module, the diversity is further improved as various spatial cues are extracted from each frame. Moreover, by modeling spatial dependencies, our model further enlarges the difference between features across scales.

**Analysis of Spatial Grouping:** We make a qualitative analysis to the grouping process of our proposed Spatial Grouping Module (SGM). Specifically, we use t-SNE [104] for visualizing feature distributions of spatial features, and we adopt the model trained without our SGM as the baseline (adopts average pooling to extract an integrated feature for each frame) for comparison. Also, we use the Davies-Bouldin Index[1] as a quantitative metric to measure the clustering performance, i.e., a lower value of the Davies-Bouldin Index indicates better separation between clusters. As shown by the qualitative and quantitative results in Figure 5, our SGM extracts spatial features with better cluster separation than the baseline, which is attributed to that our SGM enhances the distinction between features in different spatial groups. These results indicate that our proposed spatial grouping process forces the model to learn features encoding more different information. In the supplemental material, we also show Grad-CAM examples to qualitatively compare our SGM with the baseline.

---

[1]The Davies-Bouldin Index [105] measures a ratio between the intra-cluster distance and inter-cluster distance.

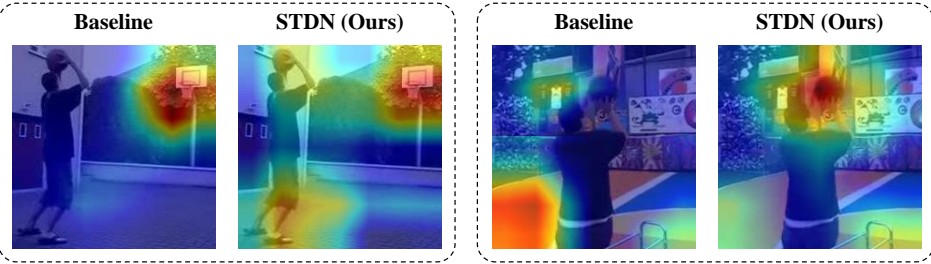

| Baseline | STDN (Ours) | | Baseline | STDN (Ours) |
| *Source Domain (Training)* | | | *Target Domain (Test)* | |

Figure 6: Grad-CAM visualization on UCF-HMDB. As shown in the figure, compared with the baseline, our proposed STDN captures more diverse class-correlated cues in the source domain, *i.e.*, including domain-specific backboards and domain-invariant basketballs. As a result, our proposed STDN generalizes better in the target domain, where backboards are invisible and thus our STDN uses the basketball for recognition instead. Best viewed in color.

**Grad-CAM Visualization:** We compare our proposed STDN with a TRN [6] model (the baseline) by Grad-CAM [106]. As shown in Figure 6, the baseline prefers to use the domain-specific backboard for recognition, which causes recognition errors in the target domains as backboards are invisible. In contrast to the baseline, our proposed STDN perceives more diverse class-correlated cues from the source domain, including some domain-invariant cues such as basketballs. As a result, our STDN can predict the correct video class by recognizing the basketball in the target video. These results demonstrate that, our proposed diversity-based approach can discover some potential domain-invariant cues, which alleviates the overfitting to domain-specific cues and leads to better generalization in the target domain.

## 5   Conclusion

In this work, we propose to explore spatial-temporal diversity to address the video domain generalization task. Our proposed Spatial-Temporal Diversification Network learns diverse spatial-temporal features in videos, which discovers potential domain-invariant cues and thus alleviates the heavy reliance on domain-specific cues. We conduct extensive quantitative and qualitative experiments on three benchmarks (including two newly designed benchmarks), and the results demonstrate the effectiveness and versatility of our approach.

**Acknowledgements.** This work was supported partially by the NSFC (U21A20471,U1911401), Guangdong NSF Project (No. 2023B1515040025, 2020B1515120085). The authors would like to thank Zhilin Zhao, Yi-Xing Peng, and Yu-Ming Tang for their valuable suggestions on model design or writing.

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
