# Supplementary Material for "Diversifying Spatial-Temporal Perception for Video Domain Generalization"

**Kun-Yu Lin**[1*]      **Jia-Run Du**[1*]      **Yipeng Gao**[1]      **Jiaming Zhou**[1]
**Wei-Shi Zheng**[1,2†]
[1]School of Computer Science and Engineering, Sun Yat-sen University, China
[2]Key Laboratory of Machine Intelligence and Advanced Computing, Ministry of Education, China
{linky5,dujr6,gaoyp23,zhoujm55}@mail2.sysu.edu.cn
wszheng@ieee.org

## S1   Comparison with AVRNA

In this section, we attempt to make a performance comparison between our proposed STDN and AVRNA [1]. Since AVRNA involves the audio modality but our work focuses on the RGB modality, we implement two variants of AVRNA for comparison as follows:

- Hard Norm Alignment loss (HNA): apply the HNA loss (Eq. (4) in [1]) for normalizing video features.
- Relative Norm Alignment loss (RNA): first divide the source domain into two subdomains following previous domain adaptation works [2, 3, 4] (as the target domain is not accessible), and then apply the RNA loss (Eq. (3) of [1]) for aligning feature distributions of the two subdomains.

We implement the above two variants based on Temporal Relation Network (TRN) [5], and all of these experiments are conducted under the same augmentation setting (*i.e.*, without MixStyle). As shown in Table S1, our STDN significantly outperforms the two variants of AVRNA on UCF→HMDB, which demonstrates the effectiveness of our model. By comparing the baseline and the two variants of AVRNA, we find that it is challenging to improve video domain generalization by simply aligning two subdomains of the source domain.

Table S1: Comparison with AVRNA.

| Method | UCF→HMDB |
|---|---|
| Baseline (TRN) | 53.1 |
| AVRNA-HNA [1] | 54.3 |
| AVRNA-RNA [1] | 55.8 |
| STDN (Ours) | 58.3 |

## S2   Analysis of Diverse Features

In this section, we demonstrate more quantitative results to justify that our STDN can perceive diverse class-correlated cues. In this experiment, we evaluate a trained STDN by dropping features produced from our proposed modules, namely Spatial Grouping Module (SGM), Spatial Relation Module (SRM) and Temporal Relation Module (TRM), for justification.

First, we drop feature from a specific spatial group. The results are shown in Table S2, where STDN-$i$ denotes that the $i$-th group is dropped for each video. We find that dropping any one of the 4 spatial groups will cause performance degradation compared with the full STDN, which demonstrates that our learned diverse spatial features encode effective class-correlated information more than noise.

---

[*]Equal contributions
[†]Corresponding author

37th Conference on Neural Information Processing Systems (NeurIPS 2023).

| Table S2: Analysis of SGM. | |
|---|---|
| Method | UCF→HMDB |
| STDN-1 | 59.7 |
| STDN-2 | 59.5 |
| STDN-3 | 59.7 |
| STDN-4 | 59.1 |
| Full STDN | 60.2 |

| Table S3: Analysis of SRM. | |
|---|---|
| Method | UCF→HMDB |
| STDN-S-1 | 59.5 |
| STDN-S-2 | 59.3 |
| STDN-S-3 | 59.7 |
| Full STDN | 60.2 |

| Table S4: Analysis of TRM. | |
|---|---|
| Method | UCF→HMDB |
| STDN-T-1 | 59.2 |
| STDN-T-2 | 58.1 |
| STDN-T-3 | 59.4 |
| STDN-T-4 | 58.9 |
| Full STDN | 60.2 |

Second, we drop feature from a space scale. The results are shown in Table S3, where STDN-S-$i$ denotes that the $i$-th space scale is dropped. As shown in the table, dropping any one of the 3 space scales will cause performance degradation compared with the full STDN, which demonstrates the effectiveness of our diversity-driven spatial relation modeling. Third, we drop feature from a time scale. The results are shown in Table S4, where STDN-T-$i$ denotes that the $i$-th time scale is dropped. It is shown that dropping any one of the 4 time scales will cause performance degradation compared with the full STDN, which demonstrates the effectiveness of our diversity-driven temporal relation modeling.

## S3  Results Using Other Backbones

In our main manuscript, we conduct all experiments based on ResNet-50. In this part, in addition, we conduct experiments based on other backbones. The table shows a comparison between our STDN and two existing methods (TRN [5] and VideoDG [6]) on UCF->HMDB, using I3D [7] and ViT-B/32 [8] as backbones. Our superior performance demonstrates the effectiveness of our design.

Table S5: Comparison on other backbones.

| Backbone | I3D | ViT-B/32 |
|---|---|---|
| TRN [5] | 68.0 | 61.3 |
| VideoDG [6] | 68.7 | 61.8 |
| STDN (Ours) | 72.1 | 64.8 |

## S4  Results on Generic Video Classification

Since our proposed STDN is applicable in generic video classification, we also conduct experiments on the generic video classification task using the Something-Something-V2 dataset [9]. As shown in Table S6, our proposed STDN can obtain performance improvement on the Something-Something-V2 dataset. It should be noted that, our proposed STDN is specifically designed for video domain generalization (VDG). Our key idea is to discover diverse class-correlated cues in the source domain, such that our model can leverage different types of cues for recognition in the target domain. The set of rich and diverse class-correlated cues is more likely to include recognition cues that are invariant (shared) across the source and target domains, compared with previous models that overfit domain-specific cues. Thus, our proposed model can generalize better in the target domain. Regarding generic video classification, it is a very different task from VDG. In generic video classification, the training and test videos follow the same distribution (namely an identical domain), thus the class-correlated cues specific in the training domain would be effective in the test domain. This is why our STDN obtains much more performance improvement in VDG compared with generic video classification.

Table S6: Generic video classification results.

| Method | Something-Something-V2 |
|---|---|
| TRN [5] | 42.6 |
| STDN (Ours) | 43.2 |

## S5  Qualitative Analysis of the Spatial Grouping Module

We show attention regions of different spatial groups obtained by our proposed Spatial Grouping Module (SGM) and the baseline (adopts average pooling to extract an integrated feature). As shown in Figure S1, for different spatial groups, the baseline obtains very similar visualization results, which demonstrates that it focuses on very similar spatial cues in different groups for video classification. By contrast, our SGM focuses on different class-correlated cues in different spatial groups (*e.g.*, humans, basketball, backboard). This comparison demonstrates that our SGM can perceive different spatial cues within individual frames for recognizing videos, which improves the spatial diversity.

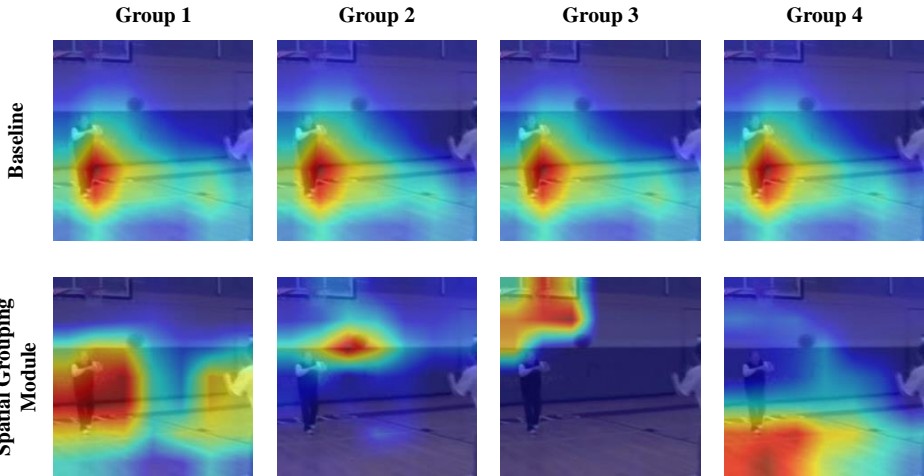

Figure S1: Attention visualization of different spatial groups by Grad-CAM on UCF-HMDB. Top: For the baseline, we partition the set of spatial features into $K$ groups by $K$-means for a fair comparison, and we use the average feature in each group for attention visualization; Bottom: For our Spatial Grouping Module (SGM), we used the $K$ integrated spatial features for attention visualization. In this experiment, we set the number of spatial groups to $K = 4$. Best viewed in color.

## S6 Hyperparameter Analysis

**The number of spatial groups $K$:** We make a quantitative analysis to the number of spatial groups obtained by our proposed Spatial Grouping Module. As shown in Figure S2, with a medium value of $K$ (*e.g.*, $K = 4$), our model obtains higher performance compared with that with a small value (*e.g.*, $K = 2$), which attributes to the improvement of spatial diversity. The performance of our model saturates as the number of spatial groups $K$ continuously gets larger (*e.g.*, $K = 8$). For a better trade-off between the effectiveness and efficiency, we set $K = 4$ by default.

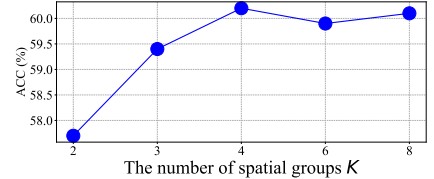

Figure S2: Analysis of the number of spatial groups $K$ on UCF→HMDB.

**The loss weight $\lambda_{\mathrm{ent}}$ for the entropy-based losses:** We make a quantitative analysis to the loss weight $\lambda_{\mathrm{ent}}$ of the entropy-based losses used in the learning of our proposed Spatial Grouping Module. As shown in Figure S3, a relatively small value of loss weight $\lambda_{\mathrm{ent}}$ (*e.g.*, $\lambda_{\mathrm{ent}} = 0.1$ or $\lambda_{\mathrm{ent}} = 0.15$) is appropriate for our proposed spatial grouping process, where different types of spatial cues are well extracted. As the loss weight $\lambda_{\mathrm{ent}}$ gets larger (*i.e.*, $\lambda_{\mathrm{ent}} \geq 0.3$), the performance of our model gradually gets lower. This is probably because that a large value of $\lambda_{\mathrm{ent}}$ disrupts the distribution of spatial features within individual frames.

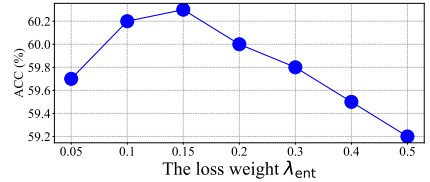

Figure S3: Analysis of the loss weight $\lambda_{\mathrm{ent}}$ on UCF→HMDB.

**The loss weight $\lambda_{\mathrm{rel}}$ for the relation discrimination loss:** We make a quantitative analysis to the loss weight $\lambda_{\mathrm{rel}}$ of our proposed relation discrimination loss, which ensures the diversity of temporal relation features learned by our Spatial-Temporal Relation Module. As shown in Figure S4, a medium value of $\lambda_{\mathrm{rel}}$ (*e.g.*, $\lambda_{\mathrm{rel}} = 0.5$) is more appropriate than a small value (*e.g.*, $\lambda_{\mathrm{rel}} = 0.1$), which makes the temporal relation features discriminative across different time scales. With a large value of $\lambda_{\mathrm{rel}}$ (*e.g.*, $\lambda_{\mathrm{rel}} = 0.9$), the performance of our model drops, which may caused by a loss of common information between temporal relation features across time scales.

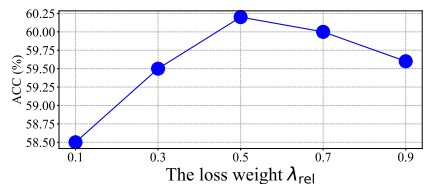

Figure S4: Analysis of the loss weight $\lambda_{\mathrm{rel}}$ on UCF→HMDB.

## S7 Discussion about Temporal Relation Network (TRN)

Although our proposed Spatial-Temporal Relation Module (STRM) is founded on TRN [5], they are designed for different tasks with different motivations. Our STRM is designed for video domain generalization, while TRN is for generic video classification. Specifically, our key idea is to explicitly perceive diverse class-correlated cues in videos by modeling multi-scale spatial-temporal dependencies, aiming to alleviate the overfitting of domain-specific cues in the source domain and make the model generalize better in unseen test domains. By contrast, TRN is a multi-scale temporal modeling module that does not address the unique challenges of VDG.

In our STRM, we contribute two novel technical designs for perceiving diverse cues: 1) we model multi-scale spatial dependencies within individual frames, which enriches the spatial diversity; 2) we propose a relation discrimination loss to constrain that different temporal dependency modeling functions capture different temporal cues, which enriches the temporal diversity. By enriching the diversity in both space and time dimensions, our STRM discovers diverse class-correlated cues from source videos for effective generalization to unseen target domains.

## S8 Broader Impacts and Limitations

In this work, we study the video domain generalization task, which aims to learn generalizable video classification models for unseen target domains by training in a source domain. Our work proposes a new video domain generalization method by exploring spatial-temporal diversity for video data. Our approach is purely algorithmic and is applicable to different types of video domain generalization scenarios (benchmarks/datasets). Therefore, it does not change the societal impacts of the video domain generalization task or the original datasets. It should be noted that, due to the task definition, the behaviors of our approach are still far from being predictable in unseen target domains, and thus users should spend care monitoring the target distributions when using the method.

One limitation of our work is that we do not consider the multi-modality nature of video data, *i.e.*, a video contains information of different modalities such as RGB, optical flow and audio. This is not the focus of our work, and we will explore this in our future works. Another limitation is a lack of analysis to our method in different types of domain shifts. It is an interesting question that how the performance of video domain generalization methods varies as the domain shift increases. However, it is very challenging to analyze these using existing video domain generalization benchmarks. We will attempt to construct new benchmarks and quantitatively analyze this problem in our future works.