# OpenReview forum: "Diversifying Spatial-Temporal Perception for Video Domain Generalization"
_NeurIPS.cc/2023/Conference — NeurIPS 2023 poster_

### Official Review · Reviewer_SFFk · 2023-07-03

**Soundness:** 3 good
**Presentation:** 3 good
**Contribution:** 2 fair
**Rating:** 5
**Confidence:** 4

**Summary:**

In this work, the authors propose a Spatial-Temporal Diversification Network (STDN) for video domain generalization. First, they intrdouce a spatial grouping method to summarize the spatial clues in each frame. Then,  they further build up spatial-temporal relations in a multi-scale manner. Finally, they show the effectiveness of the method via different experiments of video domain generalization.

**Strengths:**

1. The domain generalization problem is important for video understanding in practice.
2. The paper is written in a good structure, basically.
3. The experiments somehow show the effectiveness of the design.

**Weaknesses:**

1 Design.

I am not quite convincing by the proposed design in the paper. Bascially, the spatial grouping (or clustering) or spatial-temporal relation modeling are not particularly designed for domain generalization. It could be used for the traditional video classification problem without any difficulty. Why are these designs important for domain generalization?

2 Experiment.

2.1 The setting is not quite challenging, actually. The data sets bascially belong to the same domain. It would be interesting to see the cross domain setting, like action recognition from dark videos in UG2+ Challenge.

2.2 It would be interesting to show the results of traditional video classification setting on the popular benchmarks, like Kinetics400 or Something-Something V1 or V2.

**Questions:**

Please see the weakness section.

**Limitations:**

Please see the weakness section.

---

> ### Author Rebuttal · Authors · 2023-08-09
>
> #### **Q1. Why the proposed diversity-based modeling is important for video domain generalization (VDG)?**
>
> RE: Our model is designed for VDG, although it is applicable to traditional video classification. The core idea of our proposed designs, including our proposed Spatial Grouping Module and Spatial-Temporal Relation Module, is to perceive diverse class-correlated cues in videos. A detailed illustration to answer the question is as follows:
>
> (1) As illustrated in our manuscript (Line 31-32, Figure 1), previous video classification models usually suffer from the overfitting of some domain-specific cues in the source domain (these domain-specific cues are easy-to-fit, e.g., class-correlated contexts, as demonstrated by [19, 20, 21]). In VDG, videos from the source and target domains follow different distributions, which means that some class-correlated cues in the source domain may be unseen or not correlated with categories in the target domain. As a result, a model that overfits to the cues specific in the source domain would fail to generalize in the target domain.
>
> (2) Since the target domain is unseen, it is impossible to explicitly learn the invariant cues across the source and target domains from the data. Thus, we propose an alternative approach.
>
> (3) We propose to discover diverse class-correlated cues in the source domain, such that our model can leverage different types of cues for recognition in the target domain. The set of rich and diverse class-correlated cues is more likely to include recognition cues that are invariant (shared) across the source and target domains, compared with previous models that overfit domain-specific cues (as discuss in (1)). Thus, our proposed model can generalize better in the target domain.
>
> Regarding traditional video classification, it is a very different task from VDG. In traditional video classification, the training and test videos follow the same distribution (namely an identical domain), thus the class-correlated cues specific in the training domain would be effective in the test domain (this is very different from VDG). In our reply to Q3, we quantitatively demonstrate that our STDN can effectively generalize to videos from unseen test domains and perform well in traditional video classification.
>
> #### **Q2. Experiments on dark videos**
>
> RE: Our proposed STDN effectively generalizes to dark videos by learning from normal videos. Specifically, we conduct experiments on the challenging HMDB->ARID benchmark following ACAN [R5] (ARID is the dataset from UG2+ challenge which consists of dark videos). On HMDB->ARID, methods are implemented using the I3D backbone, and these experiments are conducted using the same augmentation setting. As shown by the table below, our proposed STDN outperforms VideoDG [13] (previous SOTA) on HMDB->ARID, which demonstrates the effectiveness of our model.
>
> | Baseline (TRN [6]) | VideoDG [13] (SOTA) | STDN (Ours) |
> | :-: | :-: | :-: |
> | 41.1 | 41.3 | 44.4 |
>
> [R5] Xu et al.: Aligning Correlation Information for Domain Adaptation in Action Recognition. TNNLS 2023.
>
> #### **Q3. Results on traditional video classification**
>
> RE: The results of our proposed STDN vs. TRN (the network that our model is founded on) on the validation set of Something-Something-v2 (SSv2) are given in the table below. We use the same training recipe for both methods (e.g., resnet50 backbone, 5 segments, 80 epoch). As shown in the table, our STDN can obtain improvement over TRN in traditional video classification. Together with our superior performance on video domain generalization (Table 1&2 in our main manuscript), we demonstrate that our STDN can effectively generalize to videos from unseen test domains and perform well in traditional video classification.
>
> | Baseline (TRN [6]) | STDN (Ours) |
> | :-: | :-: |
> | 42.6 | 43.2 |

---

> > ### Comment · Reviewer_SFFk · 2023-08-16
> >
> > Thanks for the feedback. The Rebuttal addresses my main concerns. I change my rating to borderline accept.

---

> > > ### Author Response · Authors · 2023-08-16
> > >
> > > Thank you for your time and efforts. We are encouraged by your recognition.

---

### Official Review · Reviewer_YkHU · 2023-07-06

**Soundness:** 3 good
**Presentation:** 3 good
**Contribution:** 3 good
**Rating:** 6
**Confidence:** 4

**Summary:**

In this manuscript, the authors proposed a novel Spatial-Temporal Diversification Network (STDN) for video domain generalization.  More precisely, the proposed method introduces the Spatial Grouping Module and Spatial=Temporal Relation Module to discover various groups of spatial cues within individual frames and to model spatial-temporal dependencies. Experimental results on three benchmarks show the effectiveness of the proposed method.

**Strengths:**

This paper is well-written and well-organized.

The proposed method achieves state-of-the-art results across three benchmarks.

The proposed method is straightforward and interesting.

**Weaknesses:**

The Temporal Relation Module is derived from the work cited as [6]. It would be beneficial for the authors to acknowledge this in their manuscript.

In my opinion, the proposed spatial grounding module is similar to spatial attention and the KNN model. The authors are suggested to conduct an ablation study to compare these methods for a more thorough analysis.

The proposed method is founded on Temporal Segment Networks TSN). However, the comparative methods use ensembles with various backbones. It would be advisable for the authors to replicate this approach for a fairer comparison.


**Questions:**

N.A.

**Limitations:**

N.A.

---

> ### Author Rebuttal · Authors · 2023-08-09
>
> #### **Q1. Discussion about TRN [6]**
>
> RE: Thank you, we will add discussion about this.
>
> Although our proposed Spatial-Temporal Relation Module (STRM) is founded on TRN, they are designed for different tasks with different motivations. Our STRM is designed for video domain generalization, while TRN is for generic video classification. Specifically, our key idea is to explicitly perceive diverse class-correlated cues in videos by modeling multi-scale spatial-temporal dependencies, aiming to alleviate the overfitting of domain-specific cues in the source domain and make the model generalize better in unseen test domains. By contrast, TRN is a multi-scale temporal modeling module that does not address the unique challenges of VDG.
>
> In our STRM, we contribute two novel technical designs for perceiving diverse cues: 1) we model multi-scale spatial dependencies within individual frames, which enriches the spatial diversity; 2) we propose a relation discrimination loss to constrain that different temporal dependency modeling functions capture different temporal cues, which enriches the temporal diversity. By enriching the diversity in both space and time dimensions, our STRM discovers diverse class-correlated cues from source videos for effective generalization to unseen target domains.
>
> #### **Q2. Analysis of our proposed Spatial Grouping Module (SGM)**
>
> RE: We conduct further comparison experiments to demonstrate the effectiveness of our SGM. We compare our SGM with two methods as follows:
>
> (1)	Spatial Attention (SA): following CBAM [R4], we use $K$ spatial attention modules on top of the backbone to extract $K$ features for subsequent feature modeling.
>
> (2)	$K$-means Grouping (KG): we use $K$-means algorithm to partition spatial features into $K$ groups, and then the $K$ cluster centers are used as features for subsequent feature modeling.
>
> We implement the two methods with our proposed Spatial-Temporal Relation Module (STRM). The results on UCF->HMDB are given in the table below, which shows that our SGM outperforms the two comparative methods. Our superiority is attributed to that we force the network to learn different types of features according to the guidance of our entropy-based losses, which constrain the distinction between different spatial groups.
>
> | STRM+SA | STRM+KG | STRM+SGM (Ours) |
> | :---: | :---: | :---: |
> | 56.0 | 55.5 | 58.3 |
>
> [R4] Woo et al.: CBAM: Convolutional Block Attention Module. ECCV 2018.
>
> #### **Q3. Comparison with methods using various backbones**
>
> RE: We would like to clarify that our comparison is fair, since we use ResNet50 as the backbone for both comparative methods and our STDN. In order to compare with general domain generalization methods in video domain generalization (VDG), we combine these methods with classical temporal modeling modules (i.e., TSN [5], TRN [6], TSM [7] and APN [13]) to adapt video data. Our proposed STDN has temporal modeling capabilities, and thus extra temporal modeling modules are needless. All comparative methods and our STDN is based on TSN, i.e., using the sparse temporal sampling strategy.
>
> In addition, we conduct experiments based on other backbones. The following table shows a comparison between our STDN and two existing methods (TRN and VideoDG) on UCF->HMDB. Our superior performance demonstrates the effectiveness of our design.
>
> | Backbone | TRN | VideoDG [13] (SOTA) | STDN (Ours) |
> | :---: | :---: | :---: | :---: |
> | ViT-B/32 | 61.3 | 61.8 | 64.8 |
> | I3D | 68.0 | 68.7 | 72.1

---

> > ### Comment · Reviewer_YkHU · 2023-08-15
> > **Offical comments by Reviewer YkHU**
> >
> > Thanks for the response. However, I still have some concerns about STRM. From my perspective, I can not see the specific design in STRM for the generalization.

---

> > > ### Author Response · Authors · 2023-08-16
> > >
> > > Thank you for your comment. We would like to present a detailed analysis to clarify that our STRM is a specifically designed module for video domain generalization.
> > >
> > > Our key idea is to perceive diverse spatial-temporal cues, which is critical and specific to video domain generalization (line 42-49), since it alleviates the overfitting of domain-specific cues in the source domain. Thus, our proposed STRM is diversity-driven, which enriches the feature diversity in both space and time dimensions.
> > >
> > > **(1)** Time dimension: We extract diverse temporal relation features between frames by explicit dependency modeling at multiple time scales. *More importantly*, we propose a relation discrimination loss to ensure the diversity of temporal relation features, i.e., it constrains the discrimination between temporal relation features across different scales (line 206-218). As shown in the table below, our TRM without the relation discrimination loss $L_{rel}$ has a low value of normalized MSE (i.e., low feature diversity). By contrast, introducing the loss $L_{rel}$ obtains significant improvement in terms of both normalized MSE and ACC. The normalized MSE quantitatively measures the feature diversity by the difference between temporal relation features across different time scales (please refer to Figure 5 for more analysis).
> > >
> > > | UCF->HMDB | TRM w/o $L_{rel}$ | TRM (Ours) |
> > > | :-: | :-: | :-: |
> > > | ACC | 53.1 | 55.3 |
> > > | Normalized MSE | 0.0434 | 0.3329 |
> > >
> > > To further verify the effectiveness of our diverse temporal relation features, we evaluate a trained STDN (4 time scales) by dropping features of a specific time scale (STDN-T-$i$ denotes that the $i$-th time scale is dropped for each video). As shown in the table below (UCF->HMDB), dropping any one of the 4 time scales will cause performance degradation compared with the full STDN, which demonstrates the effectiveness of our diversity-driven temporal relation modeling.
> > >
> > > | STDN-T-1 | STDN-T-2 | STDN-T-3 | STDN-T-4 | Full STDN |
> > > | :-: | :-: | :-: | :-: | :-: |
> > > | 59.2 | 58.1 | 59.4 | 58.9 | 60.2 |
> > >
> > > **(2)** Space dimension: To unleash the diversity in the space dimension, we extract diverse spatial relation features within individual frames by explicit dependency modeling at multiple space scales. As shown in the table below, our full STRM outperforms our STRM without spatial relation modeling in terms of both ACC and normalized MSE. This demonstrates that our proposed spatial relation modeling can enrich the feature diversity and improve the generalization performance. Here SGM denotes our Spatial Grouping Module.
> > >
> > > | UCF->HMDB | SGM+TRM | SGM+STRM |
> > > | :-: | :-: | :-: |
> > > | ACC | 56.7 | 58.3 |
> > > | Normalized MSE | 0. 5124 | 0.5441 |
> > >
> > > To further verify the effectiveness of our diverse spatial relation features, we evaluate a trained STDN (3 space scales) by dropping features of a specific space scale (STDN-S-$i$ denotes that the $i$-th space scale is dropped for each video). As shown in the table below (UCF->HMDB), dropping any one of the 3 space scales will cause performance degradation compared with the full STDN, which demonstrates the effectiveness of our diversity-driven spatial relation modeling.
> > >
> > > | STDN-S-1 | STDN-S-2 | STDN-S-3 | Full STDN |
> > > | :-: | :-: | :-: | :-: |
> > > | 59.5 | 59.3 | 59.7 | 60.2 |
> > >
> > > In summary, our STRM is a diversity-driven module that possesses spatial-temporal dependency modeling capability with the relation discrimination loss as guidance, thus it is specifically designed for video domain generalization.

---

> > > > ### Comment · Reviewer_YkHU · 2023-08-21
> > > > **Offical comments by Reviewer YkHU**
> > > >
> > > > Thanks for the authors' details explanation. The new explanation is more clear to me.  I am inclined to maintain my recommendation at 'Weak Accept'.

---

> > > > > ### Author Response · Authors · 2023-08-22
> > > > >
> > > > > Thank you for your time and efforts. We are encouraged that your concerns have been addressed, and we greatly appreciate your positive feedback on our work.

---

### Official Review · Reviewer_m6Ue · 2023-07-06

**Soundness:** 3 good
**Presentation:** 3 good
**Contribution:** 3 good
**Rating:** 6
**Confidence:** 4

**Summary:**

The paper addresses the problem of video domain generalization for classification task. The core idea of the paper is to enhance the diversity in class-correlated cues both in spatial and temporal dimensions with the assumption that in this diverse pool, it is more likely to capture the domain-generalizable features. To capture diversity in spatial dimensions, a spatial grouping module is proposed which forms K integrated features representing K groups of features by aggregating spatial features in each group. Two different entropy-based losses are used to ensure diversity in spatial cues. Next, the paper learns spatial relation features by sampling the integrated feature from different space scales. These spatial relation features, limited to space only, are then leveraged to learn temporal relation features. To improve the effectiveness of temporal relation features, a relation discrimination loss is also used to avoid collapse of learned temporal relation feature. The overall loss for optimisation is composed of task-specific loss, two entropy losses and a relation discrimination loss. Experiments have been conducted on three different datasets and the results claim to achieve better performance than the existing method and other image-based DG baselines.

**Strengths:**

1) Generalizing to novel domains for video modality is an important and challenging task and it carries several applications in real-world. Also, not much work has been done for video domain generalization.

2) The idea of spatial-temporal relation feature is interesting to capture the diverse class-correlated cues in search of domain-invariant features in the video data.

3) Results claim to demonstrate better performance against competing methods in all three datasets, including EPIC-kitchens-DG, UCF-HMDB, and Jester-DG.

4) Ablation studies show the clear performance contribution of different components in the proposed method.

**Weaknesses:**

1) The analyses of spatial grouping using the tSNE in Fig. 6 is not very convincing. It is not very clear how the claim that the spatial grouping does the feature group of spatial features is justified in this diagram. It is important to better understand either by visualizing or some other quantification measures that what is the clustering ability of spatial grouping mechanism.

2) It is not very clear how the diverse class-correlated cues in space and time, for which the spatial-temporal diversity module is developed,  are domain-invariant information that the paper claims to extract (L:6). Fig. 6 uses Grad-cam visualization to show the attention heatmaps but it is very general and can be applicable to any classification task.

3) In table 1, the improvement from UCF to HMDB is not very encouraging over VideoDG [13]? The paper doesn’t discuss any potential reasons for this. In fact, the the performance improvement from UCF to HMDB is less than than VideoDG [13] if Mixstyle [67], which is an off-the-shelf component,  is not used in the overall framework.

4) What is the performance of the method when only using MixStyle [67] for the datasets used in Table 3?

**Questions:**

The paper tackles the problem of generalizing video classification task to novel (unseen)domains. Overall, the paper presents interesting technical contributions aimed at capturing diverse spatial-temporal features across multiple time-scales and the results show that it the method is effective against the existing method and image-based baselines. However, there are some concerns/questions (as listed in the weaknesses 1-4) due to which my initial rating for the paper is ‘weak accept’.

Also:

What 'model selection criterion' is used to report the results?

What is the performance of the method in different types of domains shifts and how the performance varies as the domain shift increases?

What would be the performance gain compared to baseline with a different backbone, such as vision transformer-based?

**Limitations:**

The supplementary material mentions that the work doesn't consider the multi-modal nature of video data as it contains different modalities such as, RGB data, optical flow, and audio.

---

> ### Author Rebuttal · Authors · 2023-08-08
>
> #### **Q1. About the t-SNE visualization**
>
> RE: In Figure R1 (please see the PDF in the global response), we show an improved version of Figure 6, which includes a quantitative analysis of cluster separation. According to Figure R1, we qualitatively (t-SNE) and quantitatively (Davies-Bouldin Index) verify that our proposed Spatial Grouping Module (SGM) extracts spatial features with better clustering than the baseline (we use $K$-means for both methods to cluster spatial features for visualization).
> The better clustering of our SGM indicates that, the features from different spatial groups encode different spatial information.
>
> In addition, we use Grad-CAM visualization to qualitatively compare our SGM with the baseline. As shown in Figure R2, our SGM focuses on different class-correlated cues by using features from different spatial groups, while the baseline focuses on very similar regions across different groups. This demonstrates that our SGM can extract diverse class-correlated spatial cues.
>
> #### **Q2. How the proposed model extracts domain-invariant cues?**
> RE: We would like to clarify that, our proposed STDN is not an approach that explicitly extracts domain-invariant cues between the source and target domains, since target data are not accessible during training in video domain generalization (VDG). Accordingly, we propose an alternative approach for VDG. Our STDN perceives diverse class-correlated cues in the source domain, such that the model can leverage various types of cues for recognition in the target domain. The set of rich and diverse class-correlated cues is more likely to include recognition cues that are invariant (shared) across the source and target domains, compared with previous models.
>
> Our quantitative analysis in Figure 5 demonstrates that our model can improve the feature diversity, namely diverse information is encoded in our learned features.
> And, the state-of-the-art performance on three benchmarks (Table 1&2) and the ablation study (Table 3) demonstrates the better generalization in target domains, which indicates that our model effectively discovers domain-invariant cues across the source and target domains.
>
> #### **Q3. Comparison with VideoDG [13]**
>
> RE: The VideoDG method actually involves a strong data augmentation technique, i.e., Robust Adversarial Domain Augmentation (RADA). By dropping RADA from VideoDG and dropping MixStyle from our STDN, we conduct a fair comparison between our STDN and VideoDG. As shown in the table below, our STDN outperforms VideoDG on both UCF->HMDB and HMDB->UCF under the same augmentation setting.
>
> | | UCF->HMDB | HMDB->UCF |
> | :-: | :-: | :-: |
> | VideoDG w/o RADA [13] | 54.3 | 71.4 |
> | STDN w/o MixStyle (Ours) | 58.3 | 76.2 |
>
> It is an interesting question that why VideoDG obtains higher improvement over the baseline on UCF->HMDB than other benchmarks. We conjecture that, it is because the adopted RADA augmentation well simulates the distribution shift from UCF to HMDB (especially with Adversarial Pyramid Network for temporal modeling, as shown by the ablation study in Table 1 of [13]). Even though VideoDG performs well on UCF->HMDB, our STDN outperforms it. Besides, our STDN obtains superior performance on other benchmarks, which verifies the effectiveness and versatility of our design.
>
> #### **Q4. Performance with only MixStyle [37]**
>
> RE: The performance of using only MixStyle is given in the following table. As shown in the table, our proposed STDN obtains significant improvement over MixStyle, which demonstrates the effectiveness of our proposed design.
>
> | | UCF->HMDB | HMDB->UCF |
> | :-: | :-: | :-: |
> | MixStyle | 55.7 | 73.5 |
> | Full STDN (Ours) | 60.2 | 77.1 |
>
> #### **Q5. Model selection criterion**
>
> RE: Following VideoDG [13], we conduct model selection according to the validation set of the source domain.
>
> #### **Q6. Analysis of different types of domain shifts**
>
> RE: We are delighted to talk about this interesting question. As demonstrated by the experiment results, our proposed STDN obtains substantial improvement over previous SOTAs under different domain shifts, e.g., environment change (EPIC-Kitchens-DG), subclass change (Jester-DG) and large illumination shift (HMDB->ARID, as shown in the Q2 of Reviewer SFFk). These results demonstrate that our method is a promising solution for video domain generalization.
>
> Regarding the performance variation at different levels of domain shifts, it is very challenging to analyze this by using existing video domain generalization benchmarks. We thank for your valuable idea for our future works, and we will attempt to quantitatively analyze this problem in the future (e.g., by constructing new benchmarks).
>
> #### **Q7. Results based on other backbones**
>
> RE: We conduct a comparison with TRN [6] and VideoDG [13] (previous SOTA) on UCF->HMDB using ViT-B/32 and I3D as backbones. As shown in the table below, our proposed STDN outperforms VideoDG based on both backbones, which demonstrates the effectiveness of our proposed STDN.
>
> | Backbone | TRN [6] | VideoDG [13] (SOTA) | STDN (Ours) |
> | :-: | :-: | :-: | :-: |
> |ViT-B/32 | 61.3 | 61.8 | 64.8 |
> | I3D | 68.0 | 68.7 | 72.1 |

---

> > ### Comment · Reviewer_m6Ue · 2023-08-11
> >
> > I have gone through the rebuttal and other reviews. Authors have provided adequate responses to most of questions, including comparisons with VideoDG[13], performance with only Mixstyle[37] and clarification on model selection criterion. It would be better to include some clear examples in the response to Q2. Authors are strongly encouraged to include rebuttal responses in the main draft of the paper. I would be inclined toward accepting this paper and would like to hear the opinion of fellow reviewers on the rebuttal.

---

> > > ### Author Response · Authors · 2023-08-11
> > >
> > > Thanks for your recognition. We are delighted and encouraged that our responses have solved most of your concerns.
> > >
> > > We would like to provide you an intuitive explanation about our idea using the following example. Suppose people always play football in professional fields in the source domain, previous models prefer to recognize the action by the static fields. However, when people play football in a basketball court in the target domain, then those models would not recognize the action. To address this, our work proposes to capture rich and diverse cues, e.g., the fields and the act of kicking a ball (invariant across the two domains), leading to effective recognition in unseen target domains.
> > >
> > > In addition to Figure 4, we show an extra example in Figure R2 (the PDF in the global response), which demonstrates that our model captures different spatial cues by different spatial groups separately. We will provide more clear examples and also include rebuttal reposes in our main manuscript.

---

### Official Review · Reviewer_JaYX · 2023-07-07

**Soundness:** 3 good
**Presentation:** 3 good
**Contribution:** 3 good
**Rating:** 5
**Confidence:** 4

**Summary:**

The paper proposes Spatial-Temporal Diversification Network (STDN) for Video Domain Generalization (VDN). VDN is a new problem which is similar to video domain adaptation, but more challenging due to no unlabeled videos from target domain is provided. STDN is mainly designed into two modules: Spatial Grouping (soft-clustering) and Spatial-Temporal Relation (similar idea as TRN [6]). Experiments are done on 3 different benchmarks: UCF-HMDB, EPIC-Kitchens-DG, Jester-DG with good improvements over baselines. Written presentation is clear and mostly easy to read.

**Strengths:**

* The motivation of the proposed method is clearly presented and experimental results are solid, i.e., good improvements over baselines.
* Various ablations, qualitative analysis provide better understanding about the proposed method.
* Written presentation is clear and easy to read and understand.

**Weaknesses:**

* Missing a direct comparison with [14], even though [14] may use additional audio modality, an attempt to compare with [14] may make the paper more solid.

* Although the problem of general domain generalization has been studied recently, the video domain generalization is less explored, which can be either good (this paper and a few other [13,14] are among the first ones) or bad (the problem is too small with limited impact).


**Questions:**

- Given the proposed method works well on VDG, it is convinced that STDN can learn diverse set of spatial and temporal features. It is natural to ask if STDN also work on generic video classification problems e.g, on Kinetics, Something-Something-v2? It will be great if it work, if not, where it falls short, it may give further insights for video understanding?

- The Spatial-Temporal Relation Module shares some similarity with TRN [6], it would be nice to have a few sentences to compare and contrast.

* minor comments:
- citation format needs further correction, e.g., [77]

**Limitations:**

The reviewer does not foresee any potential negative social impact of this work.

---

> ### Author Rebuttal · Authors · 2023-08-09
>
> #### **Q1. Comparison with AVRNA [14]**
>
> RE: We conduct experiments to compare our STDN with AVRNA on UCF->HMDB. Since AVRNA involves the audio modality but our work focuses on the RGB modality, we implement two variants of AVRNA for comparison as follows:
>
> (1) Hard Norm Alignment loss (HNA): apply the HNA loss (Eq. (4) of [14]) for normalizing video features.
>
> (2) Relative Norm Alignment loss (RNA): first divide the source domain into two subdomains following [R2, R3] (as the target domain is not accessible), and then apply the RNA loss (Eq. (3) of [14]) for aligning feature distributions of the two subdomains.
>
> We implement these two variants based on TRN, and all of these experiments are conducted under the same augmentation setting. As shown by the table below, our STDN significantly outperforms the two variants of AVRNA on UCF->HMDB, which demonstrates the effectiveness of our model.
>
> | Baseline (TRN) | HNA | RNA | STDN (Ours) |
> | :-: | :-: | :-: | :-: |
> | 53.1 | 54.3 | 55.8 | 58.3 |
>
> [R2] Zhang et al.: Divide and Contrast: Source-free Domain Adaptation via Adaptive Contrastive Learning. NeurIPS 2022
>
> [R3] Yang et al.: Divide to Adapt: Mitigating Confirmation Bias for Domain Adaptation of Black-Box Predictors. ICLR 2023
>
> #### **Q2. Significance of studying video domain generalization**
>
> RE: In our opinion, video domain generalization (VDG) is a critical research area to develop robust video classification models capable of effectively generalizing to unseen test domains. The setting of VDG aligns closely with real-world applications, since models would face unfamiliar scenarios in practice.
>
> Compared with the widely studied general domain generalization that focuses on image data, VDG focuses on more complex video data with an extra time dimension. Therefore, VDG would suffer from large and complex domain shifts (e.g., variations of motion, unexpected absence or misalignment of short-term snippets), which cannot be addressed by general domain generalization methods. Therefore, advanced spatial-temporal modeling methods should be developed to address VDG.
>
> Hoping to further the development of this field, our work designs two new benchmarks for evaluating VDG methods, with numerous reproduced baselines. As shown in Table 1&2, general domain generalization methods perform poorly in VDG. By contrast, our work proposes a diversity-based spatial-temporal modeling approach tailored to challenges of VDG, which achieves superior performance on all the three benchmarks.
>
> #### **Q3. Performance of generic video classification**
>
> RE: The results of our proposed STDN vs. TRN on the validation set of Something-Something-v2 (SSv2) are given in the table below. We use the same training recipe for both methods (e.g., resnet50 backbone, 5 segments, 80 epoch). As shown in the table, our STDN can obtain improvement over TRN in generic video classification. Together with our superior performance on video domain generalization (Table 1&2), we demonstrate that our STDN can effectively generalize to videos from unseen test domains and perform well in generic video classification.
>
> | Baseline (TRN [6]) | STDN (Ours) |
> | :-: | :-: |
> | 42.6 | 43.2 |
>
> #### **Q4. Discussion about TRN [6]**
>
> RE: Thank you, we will add discussion in our manuscript.
>
> Overall, our proposed Spatial-Temporal Relation Module (STRM) is different from TRN, although STRM is designed based on TRN. Our STRM is designed for video domain generalization (VDG), while TRN is for generic video classification. Specifically, our STRM proposes to perceive diverse class-correlated cues in videos by modeling multi-scale spatial-temporal dependencies, aiming to alleviate the overfitting of domain-specific cues in the source domain and make the model generalize better in unseen test domains. By contrast, TRN is a multi-scale temporal modeling module that does not address the unique challenges of VDG.
>
> Technically, our STRM differs with TRN in two aspects: 1) our STRM models multi-scale spatial dependencies within individual frames, which enriches the spatial diversity; 2) we propose a relation discrimination loss to constrain that different temporal dependency modeling functions capture different temporal cues, which enriches the temporal diversity. By enriching the diversity in both space and time dimensions, our STRM discovers diverse class-correlated cues from source videos for effective generalization to unseen test domains.
>
> #### **Q5. Citation format**
> RE: Thank you, we will improve this.

---

> > ### Comment · Reviewer_JaYX · 2023-08-15
> > **Thank you for the rebuttal**
> >
> > The rebuttal partly addressed my concerns. More specific, I appreciate the direct comparison with [14], however, I am not convinced that the difference with TRN is significant. Since I appreciate the effort the author(s) put into comparison with [14], I am leaning to vote for accepting this paper, but the rating is still borderline accept.

---

> > > ### Author Response · Authors · 2023-08-16
> > >
> > > Thank you for your recognition. We would like to present more details about our work, in order to clarify the differences between our STRM and TRN.
> > >
> > > To tackle the unique challenges of video domain generalization (line 31-32), our key idea is to perceive diverse class-correlated cues in videos. Although the design of our STRM draws valuable inspiration from the classical TRN, they have important technical differences, as summarized in the following table. Due to these differences, our STRM can perceive diverse spatial-temporal cues, whereas TRN cannot.
> > >
> > > | | Multi-scale Temporal relation | Multi-scale Spatial relation | Relation discrimination |
> > > | :-: | :-: | :-: | :-: |
> > > | TRN [6]	| $\checkmark$ | | |
> > > | STRM (Ours) | $\checkmark$ | $\checkmark$ | $\checkmark$ |
> > >
> > > We emphasize that, our STRM is a straightforward yet effective adaptation of TRN for video domain generalization, which imposes diversity-driven modeling capability into TRN. Specifically, our STRM improves upon TRN to endow it with spatial-temporal dependency modeling capability. And more importantly, our relation discrimination loss constrains the temporal relation learning process, which is critical for learning diverse features. In what follows, we illustrate the differences between our STRM and TRN in detail.
> > >
> > > **(1)** We propose the *relation discrimination loss* $L_{rel}$ to ensure the diversity of temporal relation features, i.e., it constrains the discrimination between temporal relation features across different scales (line 206-218). It is a simple yet effective loss to improve the temporal diversity for better video domain generalization. As shown in the table below, the classical TRN has a low value of normalized MSE (i.e., low feature diversity). By contrast, our TRM with the loss $L_{rel}$ significantly outperforms TRN in terms of both normalized MSE and ACC. The normalized MSE quantitatively measures the feature diversity by the difference between temporal relation features across different time scales (please refer to Figure 5 for more analysis).
> > >
> > > | UCF->HMDB|TRN [6]|TRM (Ours, TRN+$L_{rel}$)|
> > > | :-: | :-: | :-: |
> > > |ACC|53.1|55.3|
> > > |Normalized MSE|0.0434|0.3329|
> > >
> > > **(2)** To enrich the spatial diversity, we extract *spatial relation* features by explicit dependency modeling at multiple space scales, while the original TRN ignores the space dimension. As shown in the table below, our full STRM outperforms our STRM without spatial relation modeling in terms of both ACC and normalized MSE. This demonstrates that our proposed spatial relation modeling can enrich the feature diversity and improve the generalization performance.
> > >
> > > | UCF->HMDB | SGM+TRM | SGM+STRM |
> > > | :-: | :-: | :-: |
> > > | ACC | 56.7 | 58.3 |
> > > | Normalized MSE | 0. 5124 | 0.5441 |
> > >
> > > In addition, our STRM addresses *a non-trivial technical challenge* of applying our Spatial Grouping Module (SGM).
> > >
> > > - Our SGM proposes to extract various spatial cues of different types, leading to features of $K$ spatial groups as the output. How to integrate features of these $K$ spatial groups and produce an integrated frame-level feature for further temporal modeling is a non-trivial challenge.
> > >
> > > - There are some optional schemes for feature integration: a) Avg: average features over different spatial groups; b) Cat: concatenate features of different spatial groups; c) SGM-t: a modified SGM that conducts grouping over spatial features from all frames of each video. We conduct empirical analysis for these schemes on UCF->HMDB, and the results are shown in the following table.
> > >
> > > | SGM+Avg+TRM|SGM+Cat+TRM|SGM-t+TRM|SGM+STRM (Ours)|
> > > | :-: | :-: | :-: | :-: |
> > > |55.8|56.7|56.0|58.3|
> > >
> > > - Compared with these schemes, our STRM is more reasonable, and our STRM outperforms other schemes with large margins as shown in the table above.
> > >
> > > **(3)** Overall: As shown in the following table, our full STRM significantly outperforms the classical TRN on UCF->HMDB, which is attributed to our diversity-driven design (as discussed in (1) and (2) above).
> > >
> > > |SGM+TRN [6]|SGM+STRM (Ours)|
> > > | :-: | :-: |
> > > |55.1|58.3|
> > >
> > > In summary, our STRM is a straightforward yet effective approach inspired by TRN, which improves spatial-temporal diversity for effective video domain generalization.
> > >
> > > In addition to our STRM, our work has two other important contributions as follows (please refer to line 48-60 for more details):
> > >
> > > 1) We propose Spatial Grouping Module to enrich the spatial diversity by embedding a clustering-like process, which is an important technical contribution.
> > >
> > > 2) We design two new benchmarks with numerous reproduced baselines (existing works publish only a limited number of benchmarks), which will further the development of the video domain generalization field.

---

### Official Review · Reviewer_bwaF · 2023-07-08

**Soundness:** 3 good
**Presentation:** 3 good
**Contribution:** 3 good
**Rating:** 5
**Confidence:** 4

**Summary:**

The paper presents STDN, a spatio-temporal diversification network designed for domain generalization. It introduces a spatial grouping module that effectively groups features from individual frames across different spatial frames. Additionally, a spatio-temporal relation module is proposed to model spatial-temporal correlations at multiple scales. The experiments demonstrate the network's good performance on three benchmarks.

**Strengths:**

- The paper is well written and easy to follow
- The results on four setup is comprehensive (I do have concerns on the results, pleas also read the next section)


**Weaknesses:**


Regarding the methodology:
- The author asserts that domain-specific cues are crucial for achieving good generalization (L43-45). Subsequently, the paper states that spatial grouping is employed to enhance the diversity of spatial modeling. It is necessary to provide further justification as to why this diversity aids in learning domain-specific features instead of introducing noise.
- The term "domain-specific feature" is frequently used in this paper; however, it is never explicitly defined. Moreover, the results fail to substantiate that the proposed method effectively learns these domain-specific features. Taking the basketball example into consideration, the backboard can be regarded as the domain-specific feature within the training set due to its construction, and the basketball itself might not be the case as you can kick the basketball, so what is the domain specific feature? should it be data driven or manually defined? When the paper claims that the proposed method can learn more representative features, supporting evidence must be provided. Currently, aside from visualization (which will be discussed later), there is a lack of evidence to support the theory that the proposed method effectively learns domain-specific features.
- It is important to note that improved results do not necessarily demonstrate that the proposed method resolves the domain generalization problem. For instance, if a stronger backbone were employed, significantly better performance could be achieved under the same experimental conditions; however, this would not imply that the stronger backbone more effectively addresses the domain generalization issue. To this end, the most straight forward experiments would be comparing against the baselines use I3D backbone, as I3D learns spatio-teamporal feature without grouping and multi-scale.

Concerning the results:
- Firstly, it should be noted that the works listed for comparison in Table 1 are incomplete. The authors could easily find numerous works on UCF-HMDB that exhibit considerably better performance.
- Given the substantial emphasis placed on spatial modeling and spatio-temporal modeling, it is crucial to compare the proposed method against previous works that utilize 3D backbones, such as CoMiX (refer to https://arxiv.org/pdf/2110.15128.pdf), which demonstrates superior performance. Additionally, it is important to include the set of baselines cited by this paper and establish a fair comparison, such as using ResNet101 as the backbone.
- It is worth pointing out that there may be instances where Grad-CAM highlights the "domain-specific" features, yet the network makes incorrect classifications. Thus, Grad-CAM alone cannot serve as conclusive evidence for improved domain generalization.

**Questions:**

Please see my comments above.

---

> ### Author Rebuttal · Authors · 2023-08-09
>
> #### **Overall clarification of our idea**
>
> We would like to clarify that, our key idea is to discover diverse class-correlated cues in videos, aiming to *alleviate the overfitting of domain-specific cues* in the source domain, NOT focus on learning domain-specific cues, as stated in our introduction (Line 31-32, 43-48). As diverse class-correlated cues are discovered from the source domain, our STDN can leverage various types of cues for recognition in the target domain. Compared with previous models, the set of diverse class-correlated cues is more likely to include recognition cues that are invariant (shared) between the source and target domains, leading to better generalization in the unseen target domain.
>
> #### **Q1. Concern about noise**
>
> RE: First, as illustrated in the overall clarification, our key idea is to perceive diverse class-correlated cues (NOT focus on learning domain-specific cues) for video domain generalization.
>
> To further justify the effectiveness of diverse features learned by our Spatial Grouping Module, we evaluate a trained STDN (4 spatial groups) by dropping features from a specific spatial group (STDN-$i$ denotes that the $i$-th group is dropped for each video). As shown in the table below (UCF->HMDB), dropping any one of the 4 spatial groups will cause performance degradation compared with the full STDN, which demonstrates that our learned diverse spatial features encode effective class-correlated information more than noise.
>
> | STDN-1 | STDN-2 | STDN-3 | STDN-4 | Full STDN |
> | :-: | :-: | :-: | :-: | :-: |
> | 59.7 | 59.5 | 59.7 | 59.1 | 60.2 |
>
> #### **Q2. About the domain-specific cues**
>
> RE: First, as illustrated in the overall clarification, our proposed STDN aims to perceive diverse class-correlated cues (NOT focus on learning domain-specific cues) for video domain generalization (VDG).
>
> Second, we clarify the definition of domain-specific cues. In our work, domain-specific cues are learned from the data rather than manually defined, similar to [48, 19, 20, 21]. In principle, domain-specific cues are recognition cues that are associated with video categories in one domain (e.g., source) but do NOT have a correlation with the (identical) categories in another domain (e.g., target). Taking the EPIC-Kitchens-DG (EPIC) benchmark as an example, videos from different domains are recorded in different environments (native kitchens), and thus the domain-specific cues learned from an EPIC domain should be some cues from the environment that are statistically correlated with the video categories (e.g., specific decorations).
>
> Our work is motivated by that, previous video classification models are prone to overfit some domain-specific cues in the source domain (e.g., class-correlated contexts, as demonstrated by [19, 20, 21]), and thus this impairs the generalization performance in unseen target domains. Accordingly, we propose a diversity-based approach to tackle VDG. In Figure 5, we quantitatively demonstrate that our proposed STDN can improve the feature diversity, i.e., diverse information is encoded.
>
> #### **Q3. Experiments based on I3D backbone**
>
> RE: The results based on I3D backbone on UCF->HMDB are shown in the below table, and all methods are implemented under the same augmentation setting. As shown in the table, our proposed STDN outperforms VideoDG (previous SOTA), which demonstrates the effectiveness of our proposed design.
>
> | Baseline (TRN [6]) | VideoDG [13] (SOTA) | STDN (Ours) |
> | :-: | :-: | :-: |
> | 68.0 | 68.7 | 72.1 |
>
> #### **Q4&Q5. Incomplete comparison on UCF-HMDB (Table 1)**
>
> RE: The works listed for comparison on UCF-HMDB are complete. The works you mentioned that have better performance (e.g., CoMix [41]) are designed for video domain **adaptation** (VDA), which is a different task from video domain **generalization** (VDG). In VDA, unlabeled videos from the target domain are accessible for training, while VDG (our setting) cannot access to any target videos. A comparison between VDA and VDG methods is not fair, since VDA methods can leverage extra target videos for training and naturally lead to better performance.
>
> As for the comparison results using 3D backbone, please refer to Q3.
>
> #### **Q6. About the Grad-CAM visualization**
>
> RE: First, as illustrated in the overall clarification, our key idea is to perceive diverse class-correlated cues (NOT focus on learning domain-specific cues) for video domain generalization.
>
> The Grad-CAM visualization in Figure 4 is a qualitative analysis to intuitively illustrate that our model can discover diverse class-correlated cues. We also conduct a quantitative analysis in Figure 5, and the results show that our proposed STDN can improve the feature diversity. Furthermore, the comparison experiments on three different benchmarks (Table 1&2) and ablation study (Table 3) quantitatively demonstrate the effectiveness of our proposed diversity-based modeling. In our reply to Q1, we quantitatively demonstrate that our model learns effective class-correlated cues more than noise.

---

> > ### Comment · Reviewer_bwaF · 2023-08-16
> > **Thanks for the additional information**
> >
> > I carefully read the rebuttal and it addressed most of my concerns.
> > I am willing to raise my rating.

---

> > > ### Author Response · Authors · 2023-08-16
> > >
> > > Thank you for your time and efforts. We are encouraged by your recognition.

---

### Author Rebuttal · Authors · 2023-08-09

Thanks to all reviewers for your constructive comments. We are encouraged that the reviewers found that, our work studies an important and practical problem (Reviewer m6Ue, SFFk) with a clear motivation (Reviewer JaYX), proposes a straightforward and interesting idea (Reviewer m6Ue, YkHU), presents comprehensive and solid experimental analysis or obtains state-of-the-art performance (Reviewer bwaF, JaYX, m6Ue, YkHU) and has good writing (Reviewer bwaF, JaYX, YkHU, SFFk).

We have carefully addressed your concerns and provided detailed responses for each review.

---

### Decision · Program_Chairs · 2023-09-21

**Decision:**

Accept (poster)

**Comment:**

The topic of video domain generalization is considered of interest, the approach is sufficiently novel, and the experiments are comprehensive and convincing. The author-responses to the remaining reviewer concerns have been well received and post-rebuttal all reviewers are supportive of acceptance. The AC agrees and encourages the authors to include all the promised additions and comparative experiments in the camera-ready paper.